# GraphChain: Large Language Models for Large-scale Graph Analysis via Tool Chaining

**Chunyu Wei**[1], **Wenji Hu**[1], **Xingjia Hao**[2], **Xin Wang**[1], **Yifan Yang**[3], **Yueguo Chen**[1]
**Yang Tian**[2], **Yunhai Wang**[1]*

[1]Renmin University of China, China [2]Guangxi University, China
[3]Beijing Jiaotong University, China

`weicy15@icloud.com, 2024000991@ruc.edu.cn, haoxingjia@st.gxu.edu.cn`
`2023103702@ruc.edu.cn, 23281027@bjtu.edu.cn`
`chenyueguo@ruc.edu.cn, ytian@gxu.edu.cn, cloudseawang@gmail.com`

## Abstract

Large Language Models (LLMs) face significant limitations when applied to large-scale graphs, struggling with context constraints and inflexible reasoning. We present GraphChain, a framework that enables LLMs to analyze complex graphs through dynamic sequences of specialized tools, mimicking human exploratory intelligence. Our approach introduces two key innovations: (1) Progressive Graph Distillation, a reinforcement learning mechanism that generates optimized tool sequences balancing task relevance with information compression, and (2) Structure-aware Test-Time Adaptation, which efficiently tailors tool selection strategies to diverse graph topologies using spectral properties and lightweight adapters without costly retraining. Experiments show GraphChain significantly outperforms prior methods, enabling scalable and adaptive LLM-driven graph analysis. [2]

## 1 Introduction

Graph-structured data represents a fundamental paradigm across diverse domains, from social networks and molecular structures to knowledge bases and recommendation systems Wei et al. [2025, 2023b, 2022d]. While large language models (LLMs) have demonstrated remarkable reasoning capabilities, they encounter significant challenges when processing graph data.

Recent approaches to enhancing LLMs' graph processing capabilities have taken two primary directions. The first attempts to adapt LLMs to directly process graph structures—either through tokenization or natural language descriptions [Chai et al., 2023, Wang et al., 2023b]. However, this approach faces **Context Exhaustion**: large-scale graphs with millions of nodes and edges cannot be effectively compressed within LLMs' context limitations, making it computationally infeasible to load entire subgraphs into their context windows (Figure 1, left).

Recognizing these limitations, a second direction draws inspiration from tool learning paradigms. Approaches like Graph-ToolFormer [Zhang, 2023a] and GraphForge [Wang et al., 2024c] pioneered integrating specialized tools with LLMs for graph reasoning, enabling models to call external graph processing functions. However, these methods primarily conceptualize tool learning as text generation, relying on single-step tool invocations with textually described graph structures. This approach leads to **Reasoning Hallucination** (Figure 1, middle), as it places unrealistic demands on individual tools to provide comprehensive functionality for complex graph analysis.

---

*Corresponding author.
[2]The code is available in https://github.com/wuanjunruc/GraphChain

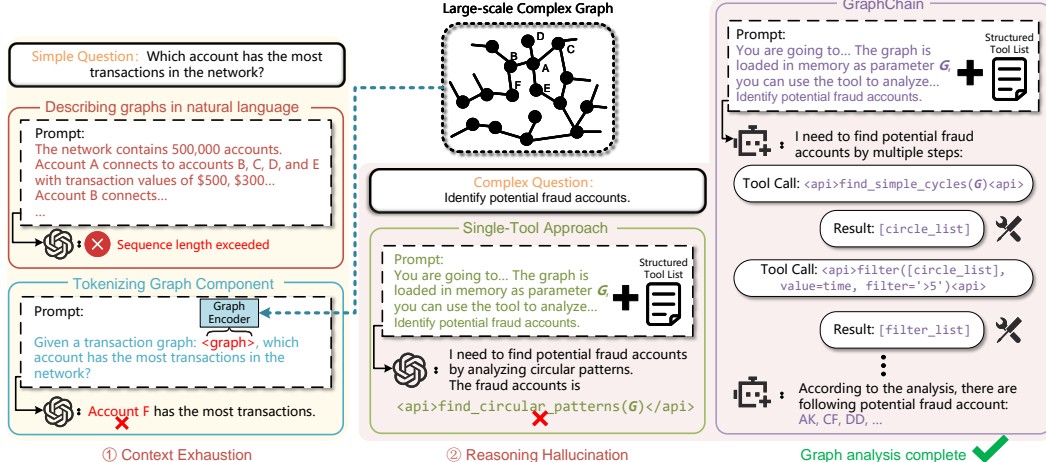

Figure 1: **Comparison of Graph Processing Approaches with LLMs.** Left: Methods suffer from Context Exhaustion where large graphs exceed LLM context windows. Center: Single-tool approaches face Reasoning Hallucination with fixed, predefined tools. Right: Our `GraphChain` framework enables human-like exploratory analysis through sequential tools that progressively narrow focus in large-scale graphs.

Complex graph analysis parallels human exploration of unknown environments. Just as humans navigate unfamiliar territories through interactive, adaptive exploration—where each step reveals information that guides subsequent decisions—effective graph analysis requires progressive, sequential information gathering rather than comprehensive analysis in one operation. A field researcher might first survey an area broadly before focusing on regions of interest; similarly, graph analysis benefits from incremental understanding built through sequential operations.

Inspired by human exploratory cognition, we propose `GraphChain`, a novel framework enabling LLMs to process large-scale graphs through dynamic tool-chaining (Figure 1, right). `GraphChain` decomposes complex graph problems into sequences of specialized operations, activating LLMs' reasoning capabilities to create, refine, and execute chains of graph processing tools. This approach allows progressive refinement and deeper exploration of graph structures, mimicking how human experts methodically investigate complex systems layer by layer.

The implementation of `GraphChain` addresses two significant technical challenges:

1. **Informative Tool Sequence Generation** requires determining optimal tool sequences for diverse analytical tasks, navigating an exponentially growing space of possible combinations. Traditional approaches struggle with this challenge due to scarce labeled data for complex graph analysis.

2. **Adaptive Graph Structure Sensing** must address real-world graph data exhibiting distributional shifts and structural variations. Unlike natural data types, graph structures are heavily human-defined with domain-specific schemas, leading to severe distribution shifts across domains.

To generate informative tool sequences, we propose a progressive graph distillation training mechanism. Our key insight is that effective graph analysis mirrors human exploration: beginning broadly and systematically narrowing focus as relevant information emerges. This approach transforms the exponential tool-selection problem into a principled information bottleneck optimization, iteratively refining both structural scope and representational complexity while preserving only task-critical information—similar to how humans selectively attend to relevant environmental cues.

For adapting to diverse graphs, we introduce a structure-aware test-time adaptation mechanism. We leverage the insight that graph topology fundamentally influences optimal analysis strategies, just as explorers adjust techniques for different terrains. Our lightweight adapter dynamically modifies tool selection policy based on spectral properties capturing essential structural characteristics, enabling `GraphChain` to maintain effectiveness across diverse graphs while preserving efficiency.

Our main contributions include:

- `GraphChain`, a novel framework leveraging Graph-Oriented Reinforcement Learning with progressive information distillation, enabling systematic exploration of large-scale graphs through interconnected tool sequences.

- A structure-aware test-time adaptation mechanism that adjusts tool-chaining strategies based on graph topology, enabling efficient transfer to diverse graph structures without costly retraining.

- Extensive experimentation demonstrating that `GraphChain` significantly outperforms existing methods by an average of 20.7%, with exceptional scalability handling graphs up to 200,000 nodes while maintaining consistent performance.

## 2 Related Work

**Tool Learning for LLMs**  Tool learning for LLMs encompasses tuning-free methods using prompting strategies like Chain-of-Thought [Wei et al., 2022e], ReAct [Yao et al., 2023], and DFSDT [Qin et al., 2023], alongside approaches integrating tools into conversations [Chen et al., 2023] or employing structured selection via graphs [Liu et al., 2024], hierarchies [Du et al., 2024], or intent filtering [Fore et al., 2024]. Meanwhile, tuning-based methods directly adapt LLM parameters [Xu et al., 2023] through behavior cloning with reinforcement learning [Qiao et al., 2024, Yu et al., 2024], fine-tuning on specialized decision data [Qin et al., 2023], frameworks for varying tool complexities [Gao et al., 2024], and self-verification mechanisms [Mekala et al., 2024].

**Graph Processing with LLMs**  Recent work enhances LLMs for graph processing via: (1) Direct processing with text or visual graph descriptions [Wang et al., 2023b, Guo et al., 2023] or specialized token sequences [Chen et al., 2024b, Wang et al., 2024b]; (2) Tool integration and agent-based methods for external function calls [Zhang, 2023b] or multi-step reasoning [Gu et al., 2024]; (3) GNN-LLM combinations using GNNs as encoders [Tang et al., 2024a] or aligning representation spaces [Su et al., 2022].

**Test-time Adaptation**  Traditional machine learning assumes identical training and testing distributions, but real-world deployments often encounter distribution shifts [Kulinski and Inouye, 2023]. Test-Time Adaptation (TTA) addresses this challenge [Liang et al., 2025, Alfarra et al., 2025]. For LLMs, adaptation techniques include test-time prompt tuning [Shu et al., 2022, Ma et al., 2023], Parameter-Efficient Fine-Tuning methods like adapters or LoRA [Hu et al., 2022] for efficient updates [Shi et al., 2024b, Muhtar et al., 2024], and "test-time compute scaling" with iterative refinement, search, or self-correction [Jaech et al., 2024, Guo et al., 2025, Suzgun et al., 2025].

## 3 Preliminaries and Problem Formulation

**Graph Notation**  Let $G = (\mathcal{V}, \mathcal{E})$ represent a graph, where $\mathcal{V} = \{v_1, v_2, \ldots, v_n\}$ is the set of $n = |\mathcal{V}|$ nodes and $\mathcal{E} \subseteq \mathcal{V} \times \mathcal{V}$ is the set of $m = |\mathcal{E}|$ edges. The adjacency matrix $\mathbf{A} \in \{0, 1\}^{n \times n}$ (or $\mathbb{R}^{n \times n}$ for weighted graphs) has entries $\mathbf{A}_{ij} = 1$ (or edge weight) if $(v_i, v_j) \in \mathcal{E}$, and 0 otherwise. Node features are represented by matrix $\mathbf{X} \in \mathbb{R}^{n \times d}$. The degree matrix $\mathbf{D}$ is diagonal with $\mathbf{D}_{ii} = \sum_{j=1}^{n} \mathbf{A}_{ij}$, and the normalized graph Laplacian is defined as $\mathbf{L} = \mathbf{I} - \mathbf{D}^{-1/2} \mathbf{A} \mathbf{D}^{-1/2}$. A node's neighborhood is $\mathcal{N}(v) = \{u \in \mathcal{V} \mid (v, u) \in \mathcal{E}\}$, and a subgraph $G' = (\mathcal{V}', \mathcal{E}')$ consists of node subset $\mathcal{V}' \subseteq \mathcal{V}$ and edge subset $\mathcal{E}' \subseteq \mathcal{E} \cap (\mathcal{V}' \times \mathcal{V}')$.

**Graph Processing Tool Library**  We define a library of graph processing tools $\mathcal{T} = \{T_1, T_2, \ldots, T_K\}$, which are implemented based on functions from the NetworkX library[3], and operate on tensor representations within the current **memory state** $\mathbf{m}$. NetworkX is a widely-used open-source Python library that provides a comprehensive set of graph processing functions, including node and edge operations, graph property calculations, and advanced analytical tools. The tools in the library are based on 45 carefully selected NetworkX functions, with details provided in Appendix E. The tools operate on tensor representations, typically containing the adjacency matrix $\mathbf{A}'$ and feature matrix $\mathbf{X}'$ for a subgraph $G' = (\mathcal{V}', \mathcal{E}')$:

$$\mathbf{m} \approx (\mathbf{A}' \in \mathbb{R}^{n' \times n'}, \mathbf{X}' \in \mathbb{R}^{n' \times d}, \ldots) \quad \text{where } n' = |\mathcal{V}'| \tag{1}$$

---

[3]https://networkx.org.

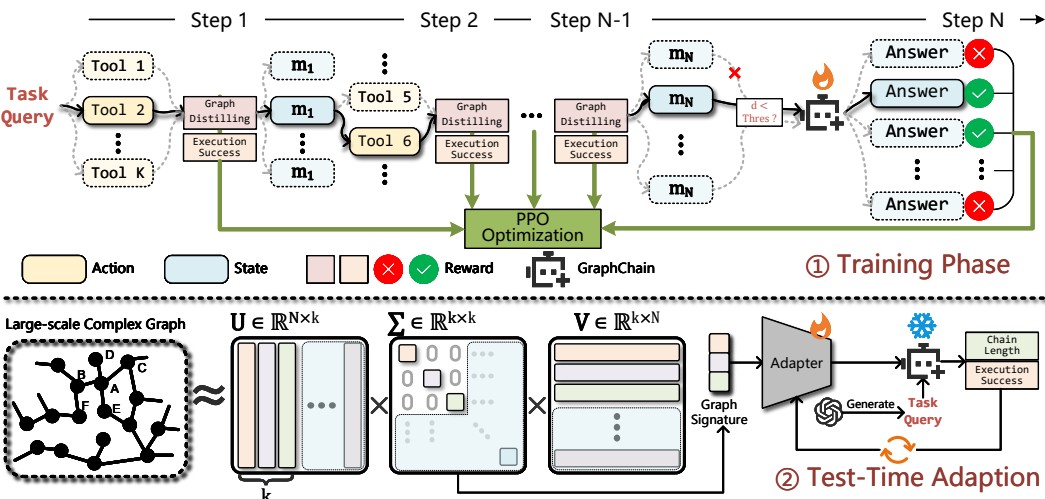

Figure 2: **(1) Training Phase:** Progressive graph distillation where the RL agent learns to select tool sequences that iteratively reduce the memory state's ($\mathbf{m}$) Graph Description Length (GDL) while maximizing task relevance. **(2) Structure-aware Test-Time Adaptation:** A lightweight adapter ($\mathcal{A}_\psi$) tuned by minimizing chain length and KL divergence generates a structure-specific soft prompt $\mathbf{P}_G$ based on the graph's SVD-derived fingerprint $\mathbf{z}_G$.

A tool $T$ takes the current memory state $\mathbf{m}$ and tool-specific parameters $\theta_T$ as input, producing two outputs: (1) A concise natural language summary $d$ of the execution outcome; (2) An updated memory state $\mathbf{m}'$. Formally, the tool function is defined as: $\quad T : (\mathbf{m}, \theta_T) \mapsto (d, \mathbf{m}')$.

This dual output mechanism allows our framework to provide context-window-friendly summaries to the LLM via $d$, while managing potentially large-scale intermediate graph data within $\mathbf{m}'$, mitigating context exhaustion when processing large graphs.

**Sequential Graph Exploration as an MDP** Given an analytical query $\mathcal{Q}$ and input graph $G$, we model sequential graph exploration as a Markov Decision Process (MDP) $M = (\mathcal{S}, \mathcal{A}, P, R, \gamma)$:

- **State Space** $\mathcal{S}$: State $s_t$ encapsulates query $\mathcal{Q}$, graph reference, action history $\{(a_i, d_i)\}_{i=0}^{t-1}$, and memory state $\mathbf{m}_{t-1}$.
- **Action Space** $\mathcal{A}$: Actions $a_t = (T, \theta_T)$ select a tool $T \in \mathcal{T}$ with parameters $\theta_T$, or 'TERMINATE'.
- **Transition Dynamics** $P$: Tool execution produces $(d_t, \mathbf{m}_t) = T(\mathbf{m}_{t-1}, \theta_T)$, updating state $s_{t+1}$ with new history and memory.
- **Reward Function** $R(s_t, a_t, s_{t+1})$: Evaluates actions based on progress and task success.
- **Discount Factor** $\gamma \in [0, 1]$: Balances immediate vs. future rewards.

The agent's policy $\pi_\theta(a_t|s_t)$, parameterized by $\theta$, generates a trajectory $\tau = \{s_1, a_1, s_2, a_2, ..., s_T, a_T\}$ representing sequential tool interactions. To maximize performance, we optimize the expected reward:

$$\nabla \overline{R_\theta} = \sum_\tau R(\tau) \nabla \pi_\theta(\tau) = \mathbb{E}_{\tau \sim \pi_\theta, (s_t, a_t) \sim \tau} \left[ R(\tau) \sum_{t=1}^{T} \nabla_\theta \log \pi_\theta(a_t|s_t) \right] \tag{2}$$

# 4 Methodology

GraphChain addresses the challenges of applying LLMs to large-scale graph analysis by formulating the problem as a sequential decision-making task solvable via reinforcement learning. Our approach centers on two core technical innovations: (1) **Progressive Graph Distillation**, which promotes informative yet compact state representations, and (2) **Structure-aware Test-Time Adaptation**, enabling dynamic adjustment to diverse graph topologies. Figure 2 provides a conceptual overview.

## 4.1 Progressive Graph Distillation

Generating effective tool sequences for complex graph queries involves navigating an exponentially large action space. To provide denser learning signals and emulate human-like analytical workflows that progress from coarse to fine, we introduce Progressive Graph Distillation.

This approach incentivizes the RL agent to pursue both the query objective and manage the complexity of its memory state $\mathbf{m}$. We train the agent to prioritize tool sequences that systematically reduce $\mathbf{m}$'s data volume while retaining task-critical information, transforming exploration into a guided search characterized by iterative refinement. The aim is to progressively shrink $\mathbf{m}$ step-by-step, eventually yielding a compact final state $\mathbf{m}_N$ suitable for direct processing within the LLM's context window.

### 4.1.1 Quantifying Memory State Volume and Relevance

Implementing progressive distillation requires quantifying two key aspects of memory state $\mathbf{m}_t$ at each step $t$: its **data volume** and its **relevance** to query $\mathcal{Q}$.

**Graph Description Length ($\mathrm{GDL}(\mathbf{m}_t)$):**  Drawing from the Minimum Description Length principle, we introduce Graph Description Length to measure the data size needed to represent the current graph state. Assuming memory state $\mathbf{m}_t$ contains subgraph $G'_t = (\mathcal{V}'_t, \mathcal{E}'_t)$ with $n'_t = |\mathcal{V}'_t|$ nodes and $m'_t = |\mathcal{E}'_t|$ edges, plus node features $\mathbf{X}'_t \in \mathbb{R}^{n'_t \times d_f}$, we define:

$$\mathrm{GDL}(\mathbf{m}_t) = L(\text{structure}) + L(\text{features}) \approx \alpha_s m'_t + \alpha_f n'_t d_f \tag{3}$$

Coefficients $\alpha_s, \alpha_f \geq 0$ weight the relative contribution of structural versus feature information.

**Task Relevance ($\mathrm{Rel}(\mathbf{m}_t, \mathcal{Q})$):**  We employ an auxiliary LLM scorer to assess the utility of $\mathbf{m}_t$ for answering query $\mathcal{Q}$. Since $\mathbf{m}_t$ may exceed the LLM's context limits, we use the concise description $d_t$ produced by the executed tool. We estimate the task relevance by:

$$\mathrm{Rel}(\mathbf{m}_t, \mathcal{Q}) \approx \mathrm{LLMScore}(\mathrm{prompt}(\mathcal{Q}, H_t, d_t)) \in [0, 1] \tag{4}$$

where $H_t = \{d_0, \ldots, d_{t-1}\}$ is the history of preceding descriptions.

### 4.1.2 Distillation-based Reward Shaping

We incorporate progressive distillation into the RL reward function $R_t = R(s_t, a_t, s_{t+1})$. The reward structure provides feedback during exploration while assessing final task completion:

$$R_t = \begin{cases} w_1 \cdot \hat{r}_t^{\mathrm{Succ}} + w_2 \cdot \hat{r}_t^{\Delta \mathrm{GDL}} + w_3 \cdot \hat{r}_t^{\Delta \mathrm{Rel}} & \text{if } t < N \\ w_{\mathrm{solve}} \cdot \mathrm{EvaluateTaskSuccess}(\mathcal{Q}, s_{N+1}) & \text{if } t = N \end{cases} \tag{5}$$

where $N$ is the final step index, and the intermediate reward components are:

- $\hat{r}_t^{\mathrm{Succ}} = \mathbb{I}(\textit{ExecutionSuccess}(a_t, s_{t+1}))$: Binary reward for valid tool execution.
- $\hat{r}_t^{\Delta \mathrm{GDL}} = \tanh\left(\beta \frac{\mathrm{GDL}(\mathbf{m}_{t-1}) - \mathrm{GDL}(\mathbf{m}_t)}{\mathrm{GDL}(\mathbf{m}_{t-1}) + \epsilon}\right) \in (-1, 1)$: Rewards reduction in relative GDL.
- $\hat{r}_t^{\Delta \mathrm{Rel}} = \mathrm{Rel}_t - \mathrm{Rel}_{t-1}$: Rewards increase in estimated task relevance.

Weights $w_1, w_2, w_3$ balance the importance of execution success, volume reduction, and relevance gain. Weight $w_{\mathrm{solve}}$ scales the final reward based on overall success in addressing query $\mathcal{Q}$.

### 4.1.3 Information Bottleneck Perspective

Our progressive distillation mechanism aligns with the Information Bottleneck principle, advocating for representations that are maximally informative about a target while being maximally compressive of input. Our reward function operationalizes this trade-off by incentivizing high task relevance while rewarding reductions in state volume.

**Proposition 4.1.** *Let the input be $X = (G, \mathcal{Q})$, containing task-relevant information $Y = \mathcal{A}_\mathcal{Q}$ (the answer) and task-irrelevant information $IR$, with the Markov structure $(Y, IR) \rightarrow X \rightarrow \mathbf{m}_t$. Assuming the relevance proxy $\mathrm{Rel}_t$ positively correlates with the mutual information $I(\mathbf{m}_t; Y)$ and the GDL serves as a complexity measure encouraging smaller $I(X; \mathbf{m}_t)$, optimizing policy $\pi_\theta$ with reward function $R_t$ guides the generation of memory states $\mathbf{m}_t$ that tend to minimize irrelevant information $I(IR; \mathbf{m}_t | Y)$ while preserving relevant information $I(\mathbf{m}_t; Y)$.*

Detailed proof is provided in Appendix A. This proposition provides theoretical support for our distillation approach. By rewarding both relevance gain and volume reduction, the RL process steers the agent toward behaviors that effectively filter graph data—reducing the representational footprint of task-irrelevant components while preserving critical information.

### 4.1.4 Policy Optimization

To optimize the LLM agent's policy $\pi_\theta$, we implement Proximal Policy Optimization (PPO), using Generalized Advantage Estimation (GAE) for improved stability:

$$\hat{A}_t^{\text{GAE}}(\theta, \omega) = \sum_{l=0}^{N-t} (\gamma\lambda)^l \delta_{t+l}, \quad \text{where} \quad \delta_t = R_{t+1} + \gamma V_\omega(s_{t+1}) - V_\omega(s_t) \tag{6}$$

Here, $\lambda \in [0, 1]$ is the GAE trace decay parameter, $V_\omega$ is the learned value function, $\gamma$ is the discount factor, and $R_{t+1}$ is the distillation-aware reward.

Following the PPO-clip approach, we maximize a clipped surrogate objective based on trajectories $\tau$ sampled from policy $\pi_\theta$:

$$\mathcal{L}^{\text{CLIP}}(\theta) = \hat{\mathbb{E}}_{\tau \sim \pi_\theta} \left[ \sum_{t=0}^{N} \min \left( \frac{\pi_\theta(a_t|s_t)}{\pi_{\theta_{\text{old}}}(a_t|s_t)} \hat{A}_t^{\text{GAE}}, \text{clip} \left( \frac{\pi_\theta(a_t|s_t)}{\pi_{\theta_{\text{old}}}(a_t|s_t)}, 1-\epsilon, 1+\epsilon \right) \hat{A}_t^{\text{GAE}} \right) \right] \tag{7}$$

where $\pi_{\theta_{\text{old}}}$ is the old policy used for generating trajectories, and $\epsilon$ is the clipping hyperparameter.

## 4.2 Structure-aware Test-Time Adaptation

### 4.2.1 Graph Structural Fingerprinting

To provide global structural awareness for large-scale graphs, we derive a concise graph fingerprint. We compute the normalized graph Laplacian $\mathbf{L} = \mathbf{I} - \mathbf{D}^{-1/2}\mathbf{A}\mathbf{D}^{-1/2}$ and consider its Singular Value Decomposition, $\mathbf{L} = \mathbf{U}\mathbf{\Sigma}\mathbf{V}^T$. The smallest singular values $\sigma_i$ capture dominant, low-frequency components reflecting macroscopic graph properties. We define the **structural fingerprint** as: $\mathbf{z}_G = (\sigma_0, \sigma_1, \ldots, \sigma_M) \in \mathbb{R}^{M+1}$.

While full SVD is intractable for very large graphs, these $M+1$ smallest singular values (where $M \ll N$) can be computed efficiently using iterative algorithms, effectively distilling essential global topology into a compact vector. We provide complexity analysis in Appendix G.

### 4.2.2 Structure-Conditioned Prompt Generation

STTA employs a continuous adaptation mechanism through adapter network $\mathcal{A}_\psi$, which maps the graph's structural fingerprint $\mathbf{z}_G$ to a soft prompt $\mathbf{P}_G = \mathcal{A}_\psi(\mathbf{z}_G) \in \mathbb{R}^{L_p \times d_{emb}}$:

This generated prompt is prepended to the standard embedding $E(s_t)$ of the agent's state, modifying the input to the frozen LLM policy: $\text{LLMInput}(s_t, G) = [\mathbf{P}_G; E(s_t)] = [\mathcal{A}_\psi(\mathbf{z}_G); E(s_t)]$.

The agent's action is then sampled from the policy conditioned on this augmented input. During adaptation, only the smaller set of adapter parameters $\psi$ are tuned, enabling efficient adaptation.

### 4.2.3 Self-Supervised Adaptation

Given the absence of ground-truth rewards for user query $\mathcal{Q}$ on unseen test graph $G_{\text{test}}$, STTA employs a self-supervised strategy using auxiliary queries. We leverage a general-purpose LLM to generate diverse auxiliary graph analysis queries relevant to $G_{\text{test}}$'s structure.

For each auxiliary query, we perform rollouts using the frozen base policy conditioned on the graph-specific prompt, yielding trajectories. The adaptation objective balances planning efficiency and policy regularization:

$$L_{\text{STTA}}(\psi) = \mathbb{E}_{\mathcal{Q}_{\text{aux},i}, \tau_i \sim \pi_\psi(\cdot|s;G_{\text{test}})} \left[ w_L N_{\tau_i} + w_{KL} \sum_{t=0}^{N_{\tau_i}-1} D_{KL}(\pi_\psi(\cdot|s_t; G_{\text{test}})||\pi_{\text{orig}}(\cdot|s_t)) \right] \tag{8}$$

The components of this objective are: (1) **Chain Length** ($N_{\tau_i}$) encouraging efficient planning, and (2) **KL Divergence Regularization** ensuring helpful but not drastic changes.

We minimize this objective using the REINFORCE algorithm, tuning $\mathcal{A}_\psi$ to generate prompts that enhance efficiency while maintaining fidelity to learned behaviors, effectively adapting the frozen policy to $G_{\text{test}}$'s specific structure.

## 5 Experiment

### 5.1 Experimental Setting

**Graph Dataset.** We evaluate `GraphChain` on five diverse graph datasets representing different real-world domains, as illustrated in Table 1.

Table 1: Statistics of graph datasets used in our experiments.

| Scenario | Dataset | #Nodes | #Edges | #Features | Type | Description |
|---|---|---|---|---|---|---|
| Citation Graphs | Cora
CiteSeer
PubMed | 2,708
3,327
19,717 | 10,556
9,104
88,648 | 1,433
3,703
500 | Directed | Academic papers connected by citation relationships [Yang et al., 2016] |
| Social Networks | Facebook
Twitter | 4,039
81,306 | 88,234
1,768,149 | -
- | Undirected
Directed | Online interactions [Leskovec and Mcauley, 2012] |
| Chemical Molecules | QM9 | ~18.0/graph | ~37.3/graph | 11 | Undirected | Molecular structures with bonds between atoms [Wu et al., 2018] |
| Traffic Networks | METR-LA | 207 | 1,515 | - | Directed | Road networks with geographic constraints [Chen et al., 2020] |
| Financial Networks | Elliptic | 203,769 | 234,355 | 165 | Directed | Transaction networks [Weber et al., 2019] |

**Instruction Data.** We constructed two complementary datasets: (1) an SFT dataset comprising 9,986 (query, tool sequence, answer) triplets based on 45 carefully selected NetworkX functions, and (2) an RL dataset containing 3,000 expert-annotated (query, answer) pairs (600 per graph scenario). We allocated 500 pairs per scenario for training and 100 for testing, with domain experts crafting exemplary instruction templates to ensure ecological validity. See Appendix F for details.

**Baselines.** We evaluated `GraphChain` against state-of-the-art methods from two categories:

(1). For Text-Instruction methods, we tested leading closed-source LLMs (`Claude-series` [Anthropic, 2024], `GPT-series` [OpenAI, 2023], and `GLM4-0520` [GLM, 2024]) using two-shot prompting with Chain-of-Thought reasoning, and reproduced specialized graph reasoning methods (`NLGraph` [Wang et al., 2023a], `GraphWiz` [Chen et al., 2024a]).

(2). For Tool-Instruction methods, we compared against recent tool-augmented approaches (`Graph-ToolFormer` [Zhang, 2023a], `GraphForge` [Wang et al., 2024c], and `ToolGen` [Wang et al., 2025]).

To ensure fair comparisons with existing baselines—all requiring the entire graph in the context window—we partitioned original graphs into subgraphs with fewer than 100 nodes for overall comparison. We use the same input for both baseline methods and `GraphForge`. In our scalability experiment (Section 5.4), `GraphChain` maintains comparable performance even when scaling to graphs with approximately 200,000 nodes. Further details are provided in Appendix D.

**Training Setup.** We used two NVIDIA A800 GPUs with LoRA-based fine-tuning (rank $r=16$, alpha=32) on the `Qwen2.5-7B-instruction` model. Further details are provided in Appendix C.

### 5.2 Main Results

Table 2 presents performance comparisons of `GraphChain` against state-of-the-art baselines, with statistical significance confirmed by two-sample t-tests ($p < 0.05$). Key insights include:

- `GraphChain` substantially outperforms all baselines, achieving 84.7% average accuracy compared to 70.2% for the best baseline (`GraphForge`), representing a 20.7% relative improvement.
- Among text-instruction baselines, `GPT-4o` with approximately 200B parameters demonstrates superior performance (59.4% average accuracy), confirming the applicability of scaling laws to graph reasoning tasks.

Table 2: Performance comparison (accuracy %) across five real-world graph reasoning scenarios.

| Model | Parameters | Financial Network | Chemical Molecule | Social Network | Citation Graph | Traffic Network | Average |
|---|---|---|---|---|---|---|---|
| **Text-Instruction Methods** | | | | | | | |
| Claude-3-Sonnet | - | $21.7 \pm 1.8$ | $47.0 \pm 2.2$ | $21.5 \pm 3.2$ | $17.7 \pm 2.1$ | $16.8 \pm 2.0$ | $24.9 \pm 2.3$ |
| GPT-3.5-turbo | ∼175B | $36.6 \pm 2.1$ | $23.0 \pm 3.7$ | $18.2 \pm 3.6$ | $12.2 \pm 0.8$ | $19.4 \pm 1.9$ | $21.9 \pm 2.4$ |
| Claude-3-Haiku | ∼20B | $12.2 \pm 3.0$ | $52.9 \pm 3.2$ | $50.3 \pm 3.4$ | $19.8 \pm 2.0$ | $13.9 \pm 2.4$ | $29.8 \pm 2.8$ |
| Claude-3-Opus | ∼137B | $23.6 \pm 2.1$ | $42.4 \pm 1.4$ | $51.9 \pm 1.3$ | $36.7 \pm 3.1$ | $43.4 \pm 3.3$ | $39.6 \pm 2.2$ |
| GraphWiz | 13B | $41.1 \pm 3.9$ | $52.4 \pm 2.6$ | $61.5 \pm 3.5$ | $68.0 \pm 2.1$ | $38.4 \pm 1.9$ | $52.3 \pm 2.9$ |
| NLGraph | ∼100B | $52.1 \pm 3.4$ | $58.4 \pm 2.5$ | $65.2 \pm 2.3$ | $59.4 \pm 0.5$ | $39.8 \pm 1.8$ | $55.0 \pm 2.1$ |
| GPT-4o | ∼200B | $57.5 \pm 1.9$ | $62.7 \pm 3.6$ | $65.2 \pm 3.9$ | $71.5 \pm 3.4$ | $43.4 \pm 1.6$ | $59.4 \pm 2.6$ |
| Claude-4-Sonnet | - | $58.2 \pm 2.1$ | $62.9 \pm 1.7$ | $61.7 \pm 4.3$ | $\underline{77.5} \pm 1.4$ | $32.8 \pm 1.9$ | $58.6 \pm 2.3$ |
| GPT-4.1 | - | $52.2 \pm 1.5$ | $63.4 \pm 2.6$ | $67.4 \pm 2.3$ | $70.0 \pm 1.9$ | $55.5 \pm 3.1$ | $61.7 \pm 2.2$ |
| Gemini-2.5-Flash | - | $25.1 \pm 1.3$ | $67.3 \pm 1.6$ | $28.1 \pm 2.1$ | $24.1 \pm 1.8$ | $24.9 \pm 1.8$ | $33.9 \pm 1.7$ |
| **Tool-Instruction Methods** | | | | | | | |
| Graph-ToolFormer | 8B | $47.5 \pm 1.9$ | $68.1 \pm 4.8$ | $74.7 \pm 4.2$ | $61.4 \pm 3.4$ | $65.8 \pm 4.5$ | $62.4 \pm 3.5$ |
| GraphForge | 8B | $63.5 \pm 3.5$ | $70.9 \pm 5.4$ | $80.4 \pm 4.2$ | $63.4 \pm 4.4$ | $73.5 \pm 3.1$ | $70.2 \pm 3.8$ |
| ToolGen | 8B | $\underline{75.8} \pm 1.1$ | $57.9 \pm 2.9$ | $79.4 \pm 2.3$ | $61.2 \pm 1.3$ | $62.7 \pm 1.5$ | $67.4 \pm 1.8$ |
| GraphChain | 7B | $\mathbf{81.5 \pm 2.2}$ | $\mathbf{81.1 \pm 0.7}$ | $\mathbf{89.6 \pm 2.0}$ | $\mathbf{83.6 \pm 2.6}$ | $\mathbf{84.1 \pm 0.3}$ | $\mathbf{84.7 \pm 1.8}$ |
| *Relative improvement (%)* | - | *+7.5%* | *+14.4%* | *+11.4%* | *+7.9%* | *+14.4%* | *+20.7%* |

† **Boldface** denotes the highest score, and underline indicates the best result among baselines.

- Specialized graph reasoning approaches like `GraphForge` (70.2% average accuracy) significantly outperform even the largest general-purpose LLMs.

- `GraphChain` achieves these results with only 7B parameters, compared to `GraphForge`'s 8B and `GPT-4o`'s 200B, demonstrating remarkable parameter efficiency.

## 5.3 Ablation Study

We introduced two variants: (1) **w/o graph distillation**, where the progressive graph distillation mechanism is disabled; and (2) **w/o test-time adaptation**, where the Structure-aware Test-Time Adaptation (STTA) component is removed during inference. Figure 3 reveals several key insights:

First, `GraphChain` consistently outperforms `GraphForge` across all graph scenarios, demonstrating the superiority of our approach. Second, Removing either component leads to performance degradation, confirming that both play critical roles in enabling effective tool-chaining and structural understanding. Third, The performance drop is more severe when graph distillation is removed compared to when disabling STTA, highlighting that progressive distillation is particularly crucial for graph analysis. Lastly, `GraphChain` without test-time adaptation still outperforms `GraphForge` in most scenarios, indicating that our multi-step tool-chaining approach with graph distillation is inherently more effective than single-step tool invocation patterns.

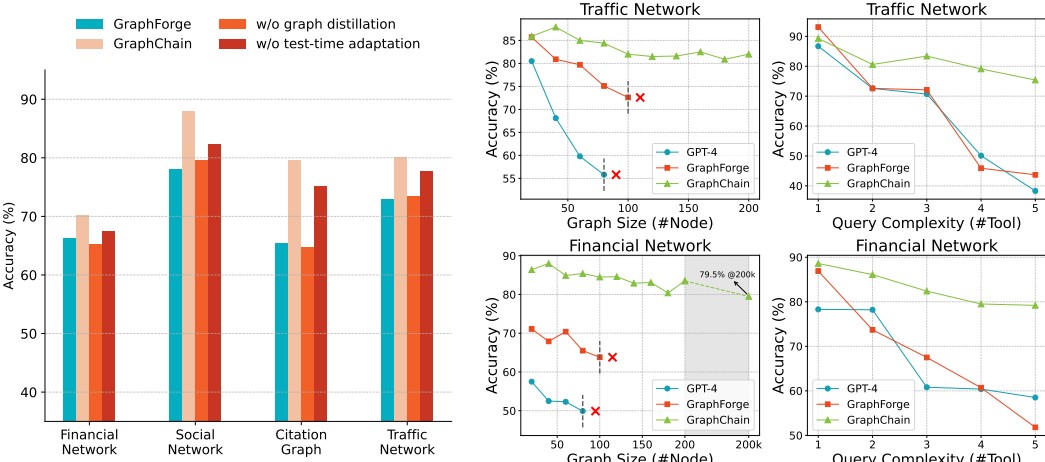

Figure 3: Impact of removing graph distillation or test-time adaptation.

Figure 4: Comparison with varying Graph Sizes and Query Complexity.

## 5.4 Scalability Analysis

We stratified our evaluation dataset based on graph size (node count) and reasoning complexity (tool sequence length) to assess how performance scales with these dimensions. Figure 4 reveals:

(1). As graph size increases, baselines exhibit significant performance degradation, with `GPT-4o` declining more dramatically, demonstrating the limitations of text-instruction for larger graphs.

(2). `GraphChain` maintains its performance advantage consistently across all graph sizes tested, including graphs with up to 200,000 nodes. This exceptional scalability stems from representing memory states through concise natural language summaries rather than direct graph descriptions.

(3). While all methods perform well on simple queries (requiring 1-2 tool calls), performance disparities increase with query complexity. Both `GPT-4o` and `GraphForge` show steep declines for queries requiring 4-5 tool calls, while `GraphChain` maintains higher accuracy, demonstrating superior capability for multi-step reasoning.

## 5.5 Transfer Learning Evaluation

To assess transfer capabilities, we fine-tuned `GraphChain` exclusively on Financial Network and evaluated on three unseen domains, comparing performance with and without the STTA module.

Table 3: Results (accuracy %) when training on Financial Network and testing on other domains.

| Model | Social Network | Citation Graph | Traffic Network |
|---|---|---|---|
| GraphChain (in-domain) | 89.6 | 83.6 | 84.1 |
| GraphChain w/ STTA | 86.8 (-3.1%) | 79.2 (-4.3%) | 80.3 (-4.5%) |
| GraphChain w/o STTA | 84.5 (-5.7%) | 75.1 (-10.2%) | 77.4 (-8.0%) |

Results in Table 3 demonstrate `GraphChain`'s strong transfer learning capabilities, with cross-domain performance closely approaching in-domain results. The STTA mechanism substantially improves transfer performance, reducing accuracy drops by 2.6%, 5.9%, and 3.5% across the three target domains compared to the variant without STTA, confirming its effectiveness in adapting to diverse graph structures without domain-specific retraining.

## 5.6 Tool Chain Analysis

To understand how `GraphChain` adapts its exploration strategy across domains, we categorized tools into six functional clusters and analyzed their usage patterns.

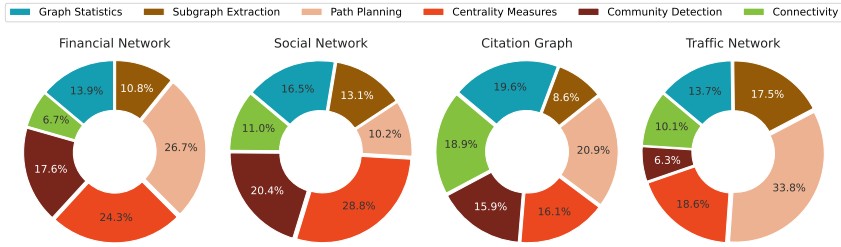

Figure 5: Distribution of tool types utilized by `GraphChain` across different graph domains.

Figure 5 reveals distinct exploration patterns adapted to each domain's characteristics. Path Planning tools dominate in Traffic Network (33.8%) and Financial Network (26.7%), reflecting the importance of traversal analysis. Social Network analysis relies on Centrality Measures (28.8%) and Community Detection (20.4%), aligning with the importance of influence and clustering. Citation Graph processing shows a more balanced distribution with significant usage of Connectivity tools (18.9%). These domain-specific variations demonstrate `GraphChain`'s ability to adaptively construct appropriate tool sequences on different graph scenarios.

Our framework is designed to be inherently robust to variations in the tool library. The core of `GraphChain` employs a reinforcement learning policy that learns to select optimal tool sequences from the available action space, rather than being hard-coded to specific tools.To empirically validate

this robustness, we conducted an additional experiment with a reduced toolset (removing 50% of tools from Centrality and Community Detection categories). Table 4 demonstrate that `GraphChain` maintains strong performance even with a reduced toolset, showcasing the adaptability of our RL agent in finding alternative tool sequences to solve tasks.

Table 4: Robustness to Tool Library Composition (accuracy %).

| Model | Financial Network | Chemical Molecule | Social Network | Citation Graph | Traffic Network | Average |
|---|---|---|---|---|---|---|
| GraphForge (Baseline) | 63.5±3.5 | 70.9±5.4 | 80.4±4.2 | 63.4±4.4 | 73.5±3.1 | 70.2±3.8 |
| GraphChain (Full Toolset) | **81.5±2.2** | **81.1±0.7** | **89.6±2.0** | **83.6±2.6** | **84.1±0.3** | **84.7±1.8** |
| GraphChain (Reduced Toolset) | 77.3±2.8 | 78.4±1.8 | 82.7±3.1 | 80.1±3.2 | 80.6±0.9 | 79.8±2.4 |

## 5.7 Robustness Study

In order to validate the robustness of `GraphChain`, we evaluated `GraphChain` with different base models. The results are shown in Table 5.

Table 5: Performance with Different Base LLMs (accuracy %)

| Base Model | Financial Network | Chemical Molecule | Social Network | Citation Graph | Traffic Network | Average |
|---|---|---|---|---|---|---|
| Qwen2.5-7B | 70.5 | 81.1 | 90.4 | 79.0 | 82.0 | 80.6 |
| Llama3.1-8B | 69.3 | 81.7 | 93.7 | 82.5 | 81.7 | 81.8 |
| GLM4-9B | 70.2 | 78.9 | 93.8 | 79.7 | 79.9 | 80.5 |

The consistent superior results across different base models demonstrate the robustness and general applicability of our approach. We also conducted supplementary experiments using Qwen2.5 models of varying sizes (3B, 7B, and 14B). The Table 6 show that `GraphChain`'s performance improves with larger model sizes. Notably, even the smaller 3B model still maintain reasonable performance under our framework.

Table 6: Comparison of Base Models with Different Sizes (accuracy %)

| Model Size | Financial Network | Chemical Molecule | Social Network | Citation Graph | Traffic Network |
|---|---|---|---|---|---|
| Qwen2.5-3B | 63.1% | 56.9% | 70.2% | 74.4% | 73.4% |
| Qwen2.5-7B | 81.5% | 81.1% | 89.6% | 83.6% | 84.1% |
| Qwen2.5-14B | 85.7% | 85.4% | 92.2% | 83.2% | 89.7% |

# 6 Conclusions and Limitation

In this paper, we introduced GraphChain, a novel framework that enables LLMs to effectively process and reason over large-scale graph data through dynamic tool-chaining. By integrating progressive graph distillation with structure-aware test-time adaptation, GraphChain addresses the fundamental challenges of context exhaustion and reasoning hallucination that plague existing graph processing approaches. Our extensive experiments across diverse domains demonstrate that GraphChain significantly outperforms prior methods.

Our current implementation primarily focuses on static graphs and may require adaptation for dynamic or temporal graph structures that evolve over time. The tool library used in our experiments, though comprehensive, could be expanded to include more domain-specific operations for specialized applications. These limitations present valuable directions for future research.

## Acknowledgments and Disclosure of Funding

This research was supported by NSFC (No.6250072448, No.62132017 and No.U2436209), the Shandong Provincial Natural Science Foundation (No.ZQ2022JQ32), the Beijing Natural Science

Foundation (L247027), the Fundamental Research Funds for the Central Universities, the Research Funds of Renmin University of China, and the Young Elite Scientists Sponsorship Program by CAST under contract No. 2022QNRC001. It was also supported by Big Data and Responsible Artificial Intelligence for National Governance, Renmin University of China.

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

# A    Proof of Proposition 4.1

We start with the fundamental assumptions:

1. The input $X$ is generated from underlying factors including task-relevant information $Y$ and task-irrelevant information $IR$.

2. The process forms a Markov chain: $(Y, IR) \rightarrow X \rightarrow \mathbf{m}_t$. This signifies that the memory state $\mathbf{m}_t$ is generated based on the input $X$, which itself is derived from the underlying factors $(Y, IR)$.

3. The optimization objective derived from the reward function $R_t$ (Eq. 5) encourages policies that produce trajectories where intermediate states $\mathbf{m}_t$ have high task relevance $\mathrm{Rel}(\mathbf{m}_t, \mathcal{Q})$ and low complexity/volume $\mathrm{GDL}(\mathbf{m}_t)$.

4. Based on the proposition's statement, maximizing relevance correlates with maximizing $I(Y; \mathbf{m}_t)$, and minimizing GDL correlates with minimizing the overall information captured from the input, $I(X; \mathbf{m}_t)$.

According to the Data Processing Inequality (DPI) [Beaudry and Renner, 2011] applied to the Markov chain $(Y, IR) \rightarrow X \rightarrow \mathbf{m}_t$, the information that the final representation $\mathbf{m}_t$ retains about the initial factors $(Y, IR)$ cannot exceed the information it retains about the intermediate variable $X$:

$$I((Y, IR); \mathbf{m}_t) \leq I(X; \mathbf{m}_t) \tag{9}$$

Now, we apply the chain rule for mutual information to the term on the left-hand side:

$$I((Y, IR); \mathbf{m}_t) = I(Y; \mathbf{m}_t) + I(IR; \mathbf{m}_t | Y) \tag{10}$$

Here, $I(Y; \mathbf{m}_t)$ represents the information that the memory state $\mathbf{m}_t$ contains about the relevant variable $Y$. The term $I(IR; \mathbf{m}_t | Y)$ represents the *additional* information that $\mathbf{m}_t$ contains about the irrelevant variable $IR$, given that the relevant information $Y$ is already known. This term quantifies the amount of irrelevant information captured by $\mathbf{m}_t$ beyond what is already explained by its correlation with $Y$.

Substituting the expansion from Eq. 10 into the DPI (Eq. 9), we obtain:

$$I(Y; \mathbf{m}_t) + I(IR; \mathbf{m}_t | Y) \leq I(X; \mathbf{m}_t) \tag{11}$$

Rearranging this inequality gives us an upper bound on the conditional mutual information involving the irrelevant component:

$$I(IR; \mathbf{m}_t | Y) \leq I(X; \mathbf{m}_t) - I(Y; \mathbf{m}_t) \tag{12}$$

This inequality shows that the amount of irrelevant information retained in $\mathbf{m}_t$ (conditioned on the relevant part $Y$) is upper-bounded by the difference between the total information $\mathbf{m}_t$ captures from the input $X$ and the useful information it captures about the target $Y$.

Now, let's consider the optimization objective implied by the progressive distillation reward function (Eq. 5). This objective aims to find a policy $\pi_\phi$ that maximizes the expected return. The reward structure encourages steps that increase relevance (proxy for $I(Y; \mathbf{m}_t)$) and decrease GDL (proxy for $I(X; \mathbf{m}_t)$). Thus, the optimization process implicitly seeks intermediate states $\mathbf{m}_t$ that achieve a trade-off, aiming for:

- High $I(Y; \mathbf{m}_t)$ (maximize information about the answer)
- Low $I(X; \mathbf{m}_t)$ (minimize total information stored, i.e., compress the input)

By striving to maximize $I(Y; \mathbf{m}_t)$ and minimize $I(X; \mathbf{m}_t)$, the optimization process is effectively attempting to minimize the upper bound $I(X; \mathbf{m}_t) - I(Y; \mathbf{m}_t)$ found in Eq. 12.

Minimizing this upper bound serves as a direct mechanism to suppress the amount of irrelevant information $I(IR; \mathbf{m}_t | Y)$ contained in the memory state $\mathbf{m}_t$. Therefore, optimizing the distillation-aware reward function encourages the policy to generate representations that form an information bottleneck, preserving task-relevant information while actively discarding task-irrelevant information, as quantified by the bound derived from the Data Processing Inequality.

**Query:**
Help me identify the largest simple cycle in this transfer graph, and then determine which transactions within this cycle have amounts exceeding 1,000,000.

**GraphChain's Analysis Process**

---- Step 1 ----

Thought:
First, clarify that the type of the graph is a multidigraph. To facilitate the subsequent cycle detection, it needs to be converted to an undirected graph, which can avoid the complexity caused by direction issues during cycle detection. Use the to_undirected function from the NetworkX library to convert the multidigraph G into an undirected graph.

Action:
`<API>to_undirected()</API>`

Result:
The result is not a dictionary object, but a graph object named undirectedgraph.

Result_Description:
undirectedgraph is the undirected graph obtained after conversion, with the direction of edges removed to facilitate subsequent cycle detection operations. You can proceed with cycle detection and other operations on undirectedgraph, such as using other functions from NetworkX to process this graph.
(The number of tokens in the result exceeds the limit of the context window at this point.)

---- Step 2 ----

Thought:
After obtaining the undirected graph, it is necessary to find all the simple cycles within the graph. The simple_cycles function is used for this purpose, which returns a list of all simple cycles in the graph, with each simple cycle represented as a list of nodes.

Action:
`<API>simple_cycles()</API>`

Result:
The variable largestcycle stores the list of the largest cycle, where the cycle is represented as a list of nodes.

Result_Description:
largestcycle contains the largest cycle in the graph, represented as a list of nodes. You can proceed with further operations on largestcycle, such as checking the weights of the edges within the cycle.
(The number of tokens in the result exceeds the limit of the context window at this point.)

---- Step 3 ----

Thought:
For the largest cycle, it is necessary to tally the weights of all edges within the cycle. The get_edge_data() function is required to obtain the weight of each edge. Subsequently, each edge in the cycle is iterated over to determine whether its weight exceeds 1,000,000.

Action:
`<API>get_edge_data()</API>`

Result:
For the largest cycle, it is necessary to tally the weights of all edges within the cycle. The get_edge_data() function is required to obtain the weight of each edge, and then determine whether the weight of an edge exceeds 1,000,000 by iterating through each edge in the cycle.

Result_Description:
filteredges is a list that stores the filtered edges. Each edge is represented as a dictionary composed of source, target, and weight. You can iterate through filteredges to output the final results.
(The number of tokens in the result is less than the limit of the context window, and the input to the large language model is in the form of text.)

---- Step 4 ----

Thought:
Through the first three steps, we have successfully extracted the largest cycle in the graph and filtered out the edges with weights exceeding 1,000,000, which are as follows:[{"source": 249, "target": 963, "weight": 6393000}, {"source": 1, "target": 92, "weight": 8975610}, {"source": 72, "target": 647, "weight": 2402500}, {"source": 31, "target": 96, "weight": 1297921}, {"source": 96, "target": 0, "weight": 2553000}]

Action:
`<API>Finish->answer</API>`

Answer: After analyzing the cycles in the given transfer graph, we identified the largest cycle and filtered it based on weight. The results are as follows: The list of edges with weights exceeding 1,000,000 is [{"source": 249, "target": 963, "weight": 6393000}, {"source": 1, "target": 92, "weight": 8975610}, {"source": 72, "target": 647, "weight": 2402500}, {"source": 31, "target": 96, "weight": 1297921}, {"source": 96, "target": 0, "weight": 2553000}].

Figure 6: A typical case of `GraphChain` on Financial Networks.

# B Case Study

We present a case study in Figure 6 involving a transmission network from the Elliptic Dataset, comprising 11,896 nodes and 31,054 edges—exceeding the context processing capabilities of current LLMs. `GraphChain` demonstrates robust capability to systematically handle this task through a series of tool invocations. The Progressive Graph Distillation mechanism enables iterative reduction of information complexity, focusing on the most salient features.

Due to context window limitations, we store intermediate processing results in memory rather than directly inputting them into the LLM. The model outputs detailed descriptions of current processing results and their usage methods to facilitate further analysis. Only when intermediate results' token count falls below the maximum context window limit are they directly input into the LLM, maximizing the model's advantages while minimizing limitations.

# C Details of Experimental Setup

We provide comprehensive details on our experimental setup to ensure reproducibility. All experiments were conducted on 2 NVIDIA A800 80GB GPUs, using LoRA-based fine-tuning (rank $r$=16, alpha=32) on the `Qwen2.5-7B-instruction` model.

## C.1 Training Configuration

Our training pipeline consisted of three main stages:

Table 7: Comparison of baseline methods and their corresponding models for graph reasoning.

| LLM Type | Open Source | Method | Base Model |
|---|---|---|---|
| Text Instruction | ✗ | Two-shot | `Claude-series` [Anthropic, 2024] |
| | ✗ | Two-shot | `GPT-series` [OpenAI, 2023] |
| | ✓ | `NLGraph` [Wang et al., 2023a] | `GPT-4-turbo` |
| | ✓ | `GraphWiz` [Chen et al., 2024a] | `Llama2-13B` |
| Tool Instruction | ✗ | Function Calling | `GPT-3.5-turbo` [OpenAI, 2023] |
| | ✗ | Function Calling | `GPT-4o` [OpenAI, 2023] |
| | ✗ | Function Calling | `GLM4-0520` [GLM, 2024] |
| | ✓ | `Graph-ToolFormer` [Zhang, 2023a] | `Llama3-8B` |
| | ✓ | `GraphForge` [Wang et al., 2024c] | `Llama3-8B` |

- **Supervised Fine-Tuning (SFT) Stage:** We used a learning rate of $5 \times 10^{-5}$ with 4% warmup and a cosine scheduler for 8 epochs. This initial phase established the model's ability to follow graph reasoning instructions.

- **Reinforcement Learning (RL) Stage:** We implemented Proximal Policy Optimization (PPO) with step-level rewards, departing from traditional RLHF approaches that apply rewards solely to the final step. Our implementation used:
    - Learning rate: $1 \times 10^{-5}$
    - Batch size: 8
    - Initial KL coefficient: 0.3
    - Loss coefficient ($\beta$): 0.15
    - GAE parameter ($\lambda$): 0.95
    - Discount factor ($\gamma$): 0.99

- **Test-Time Adaptation Stage:** For the structure-aware adaptation mechanism, we configured:
    - Learning rate: 0.01
    - Batch size: 10

### C.2 Inference Settings

During inference, we used a temperature of 0.7 and top-p value of 1.0, optimizing for a balance between diversity and coherence in the generated tool chains.

Complete configuration files and scripts are available in our code repository to facilitate reproduction of our results.

## D    Baseline Implementation

To comprehensively evaluate `GraphChain`, we implemented several state-of-the-art graph reasoning baselines spanning both Text-Instruction and Tool-Instruction paradigms. Table 7 summarizes these baseline methods and their corresponding backbone models.

### D.1    Text-Instruction Methods

For closed-source LLMs (`Claude-series`, `GPT-series`, and `GLM4-0520`), we employed two-shot Chain-of-Thought (CoT) prompting to stimulate structured reasoning. Table 8 illustrates our prompting approach with an example.

For specialized graph reasoning methods, we reproduced:

- `GraphWiz`: We reimplemented using the `Llama2-13B` backbone with a learning rate of 2e-5 for 3 epochs and maximum sequence length of 2048. During inference, temperature was set to 0 with maximum output tokens of 1024 to ensure stable generation.

Table 8: Example of two-shot prompting used for text-instruction baselines.

**Prompt Template**

*You are an AI assistant specialized in graph reasoning. For each problem, first extract the graph structure, then solve the task step by step.*

**Example 1:**

**Input:** Given a weighted directed graph with edges: `[(0, 2, {'weight': 3}), (0, 3, {'weight': 7}), (1, 0, {'weight': 2}), (1, 4, {'weight': 8}), (2, 4, {'weight': 1}), (3, 4, {'weight': 3})]`. Find the shortest path from node 1 to node 4.

**Reasoning:** First, I'll extract the graph structure.

`Graph: [(0, 2, {'weight': 3}), (0, 3, {'weight': 7}), (1, 0, {'weight': 2}), (1, 4, {'weight': 8}), (2, 4, {'weight': 1}), (3, 4, {'weight': 3})]`

Now I'll find the shortest path from node 1 to node 4.

- Direct path: 1 → 4 with weight 8
- Indirect path: 1 → 0 → 2 → 4 with weight 2 + 3 + 1 = 6
- Indirect path: 1 → 0 → 3 → 4 with weight 2 + 7 + 3 = 12

The shortest path is 1 → 0 → 2 → 4 with total weight 6.

**Answer:** The shortest path from node 1 to node 4 is 1 → 0 → 2 → 4 with total weight 6.

**Example 2:**

**Input:** Given an undirected graph with edges: `[(0, 1), (0, 3), (1, 2), (1, 4), (2, 5), (3, 4), (4, 5)]`. Determine if there exists a cycle in this graph.

**Reasoning:** Let me extract the graph structure.

`Graph: [(0, 1), (0, 3), (1, 2), (1, 4), (2, 5), (3, 4), (4, 5)]`

To check for cycles, I'll trace possible paths:

Starting from node 0: 0 → 1 → 2 → 5 → 4 → 3 → 0

This forms a cycle: 0 → 1 → 2 → 5 → 4 → 3 → 0

**Answer:** Yes, there exists a cycle in the graph.

**Now solve the following problem:**

**Input:** [Problem description]

---

- `NLGraph`: Following the original implementation, we provided 4 exemplars for connectivity and cycle tasks, and 5 exemplars for other tasks due to context size limitations. For fair comparison, we used the standardized test set across all experiments.

### D.2 Tool-Instruction Methods

We implemented tool-augmented approaches including:

- `Graph-ToolFormer`: We reimplemented this approach based on the `Llama3-8B` model using LoRA (rank $r$=16, alpha=32) with a learning rate of 1e-5 and weight decay of 1e-2 for 3 epochs. For generation, we used beam search with 5 beams, top-k of 5, top-p of 0.95, and temperature of 0.7.

- `GraphForge`: We implemented based on `Llama3-8B` using LoRA (rank $r$=16, alpha=32) with a learning rate of 5e-5 for 5 epochs. Inference settings matched our `GraphChain` configuration with temperature of 0.7 and top-p of 1.0.

- Function Calling: For closed-source models supporting function calling (`GPT-3.5-turbo`, `GPT-4o`, and `GLM4-0520`), we implemented the same graph processing functions used in `GraphChain` as external API tools, allowing these models to leverage structured tool invocation capabilities during inference.

All baseline implementations were executed using the same hardware setup as `GraphChain`: two NVIDIA A800 GPUs for fine-tuning and inference with open-source models. For closed-source models, we utilized their respective official API interfaces. To ensure fair comparison across all methods, we partitioned original graphs into subgraphs with fewer than 100 nodes for evaluation,

Table 9: NetworkX Functions Categorized by Graph Analysis Task

| Category | NetworkX Functions |
|---|---|
| Basic Graph Properties | `G.number_of_nodes()`, `G.number_of_edges()`, `G.has_node(n)`, `G.has_edge(u, v)`, `G.degree()`, `G.in_degree()`, `G.out_degree()`, `G.get_edge_data(u, v)` |
| Centrality Metrics | `nx.betweenness_centrality()`, `nx.closeness_centrality()`, `nx.degree_centrality()`, `nx.eigenvector_centrality()`, `nx.harmonic_centrality()`, `nx.percolation_centrality()`, `nx.second_order_centrality()`, `nx.subgraph_centrality()` |
| Connectivity and Components | `nx.strongly_connected_components()`, `nx.weakly_connected_components()`, `nx.articulation_points()`, `nx.bridges()`, `nx.k_edge_components()`, `nx.k_node_components()`, `nx.node_connectivity()`, `nx.edge_connectivity()` |
| Shortest Paths and Distances | `nx.all_pairs_shortest_path()`, `nx.all_pairs_shortest_path_length()`, `nx.dijkstra_path()`, `nx.dijkstra_path_length()`, `nx.floyd_warshall()` |
| Clustering and Communities | `nx.average_clustering()`, `nx.clustering()`, `nx.transitivity()`, `nx.triangles()`, `nx.label_propagation_communities()`, `nx.louvain_communities()` |
| Flow Algorithms | `nx.boykov_kolmogorov_min_cut()`, `nx.dinic_min_cut()`, `nx.edmonds_karp_min_cut()`, `nx.minimum_cut()` |
| Cycle Detection | `nx.simple_cycles()`, `nx.cycle_basis()` |
| Topological Sorting | `nx.topological_sort()`, `nx.is_directed_acyclic_graph()`, `nx.all_topological_sorts()`, `nx.topological_generations()` |

while separately testing `GraphChain`'s scalability on full-sized graphs with up to 200,000 nodes in Section 5.4.

## E    Graph Analysis Tool Library

To construct an effective graph question-answering system, we selected 45 functions from the NetworkX library through a systematic review of graph analysis tasks prevalent in academic research and practical applications. Table 9 shows the complete list of selected functions. The selection process prioritized coverage of eight core dimensions of graph analytics:

- `Basic Graph Properties` – Functions providing structural metadata, including node/edge counts, degree distributions, and adjacency queries.
- `Centrality Metrics` – Measures for node influence, spanning degree centrality to advanced methods (eigenvector, percolation, and Katz centrality).

- `Connectivity and Components` – Tools for evaluating graph robustness, such as articulation points, bridges, and strongly/weakly connected components.
- `Shortest Paths and Distances` – Algorithms for unweighted and weighted paths, critical for routing and diffusion modeling.
- `Clustering and Communities` – Modular structure analysis via clustering coefficients and detection algorithms (e.g., label propagation, Louvain).
- `Flow Algorithms` – Maximum flow and minimum cut computations using multiple methodologies (e.g., Edmonds-Karp).
- `Cycle Analysis` – Feedback loop identification in directed and undirected graphs.
- `Topological Sorting` – Dependency resolution for directed acyclic graphs (DAGs).

While not exhaustive, this set was carefully selected to balance *analytical breadth* and *computational efficiency*, ensuring system responsiveness and interpretability. Future work may integrate domain-specific or higher-order analytics, but this toolset is representative and sufficient for general-purpose graph analysis.

# F  Data Construction

This section details the creation of datasets used for training and evaluating `GraphChain`, including fine-tuning data and graph datasets across five real-world scenarios.

## F.1  Fine-tuning Dataset

We constructed a comprehensive and robust dataset for fine-tuning `GraphChain` through a systematic, multi-faceted approach:

### F.1.1  SFT Dataset Construction

We curated 45 commonly used APIs from the NetworkX library based on relevance and usage frequency in graph-related tasks. To ensure diverse instruction coverage, we employed `ChatGPT` to generate various instructions tailored to these APIs. For each iteration, we randomly sampled APIs and prompted `ChatGPT` to reverse-engineer instructions centered around them, ensuring comprehensive coverage across the API set.

To enhance practical relevance, human experts crafted three exemplar instructions for each subgroup within five distinct real-world graph scenarios. These expertly designed prompts served as high-quality references, grounding the dataset in realistic use cases.

Our structured prompting strategy guided `ChatGPT` to produce outputs in a standardized format:

$$\{Thought: \ ...Action: \ ...\}$$

Each action explicitly invoked an API with required parameters (e.g., `G.get_edge_data(8, 0, default=None)`). The outputs were fed into a code generator to produce executable code, which was then executed to obtain results formatted as:

$$\{"error": \ "...", "response": \ "..."\}$$

These results were appended to the input for subsequent steps, creating a coherent action sequence. We introduced two auxiliary functions: `Finish->answer` (signaling successful task completion) and `Finish->giveup_and_restart` (allowing model reset and retry in cases of persistent errors).

Through this pipeline, we generated 9,986 (instruction, solution path) pairs that encapsulate a wide range of API-driven tasks reflecting the complexity of real-world graph-based problem-solving. Table 11 shows an example from our SFT dataset.

### F.1.2  RL Dataset Construction

For the reinforcement learning phase, we constructed a dataset with reward values for each step. We used GPT-4 to score each step based on three dimensions:

Table 10: The prompt template for constructing the SFT dataset.

| Category | Description |
|---|---|
| Dataset Name | `Citation-Network.txt` |
| Dataset Type | `MultiDirected Graph` |
| Dataset Content | The citation data between research papers. Directed edge A to B means that paper A cites paper B. The graph construction operation is: `G = nx.MultiDiGraph(), G.add_edge(paper1, paper2)`, where `paper1` and `paper2` are research papers. String type is used to store nodes. |
| Task | Generate a complex graph problem and its step-by-step solution process. |
| Output Type | `JSON` |
| Output Rules | **(1)** The output must be a JSON containing a series of "from" and "value" as shown in the example, using English. **(2)** Provide the user problem in "value" under "user", generate the response in "value" under "assistant", and generate API return results in "value" under "function". **(3)** The output can have only this JSON data with no additional information. **(4)** Follow the format of the example but exclude the key name "example". |
| Special Attention | In the "assistant" response, provide detailed thought processes without code, using NetworkX methods. Mark called APIs with `<API>...</API>`, e.g., `<API>nx.dfs_edges(graph, source=10)</API>`. Format outputs as: `Thought: ... Action: <API>...</API>` When `<API>...</API>` is encountered, provide "function" with fabricated API results: `{"error": "", "response": ""}` Continue this process until final result. Final "assistant" format should be: `Thought: ... Action: <API>Finish->answer</API>` `Answer: ...` Or: `Thought: ...` `Action: <API>Finish->give_up_and_restart</API>` |
| Design Requirement | Design Q&A around the `{API_NUM}` APIs `{API_NAME}`. Focus on collaboration between APIs, where later APIs utilize results from previous ones for further reasoning. |

- **API Correctness**: Whether the tool invocation in the current step is valid
- **Thought and API Effectiveness**: The relevance of the tool selection to solving the query
- **Graph Distillation**: Whether the tool reduces the information content of the graph data

Table 12 shows the prompt template used for RL data construction.

Table 13 shows two examples from our RL dataset with corresponding evaluation scores.

### F.2 Graph Dataset

We extracted subgraphs from existing real-world datasets to construct graph datasets for five scenarios, each corresponding to different graph types:

- **Financial Networks**: From the Elliptic dataset, containing Bitcoin transaction graphs
- **Chemical Molecules**: From the QM9 dataset, including molecular structure graphs where atoms are nodes and chemical bonds are edges

Table 11: Example of SFT data with step-by-step reasoning and API calls.

| From | Value |
|------|-------|
| User | *How to identify the neighbor nodes of node 15 and analyze the fund flow between these neighbor nodes? Begin!* |
| Assistant | **Thought:** To identify the neighbor nodes of node 15 and analyze fund flow between them, we need to proceed step by step. First, obtain all direct neighbor nodes of node 15 and transaction information. Then analyze fund flow between these neighbors.
Let's obtain neighbor nodes and transaction information using `<API>graph.out_edges(15, data=True)</API>` and `<API>graph.in_edges(15, data=True)</API>` to get all outgoing and incoming edge information.
**Action:** `<API>graph.out_edges(15, data=True)</API>` |
| Function | `{"error": "", "response": "Direct outgoing transaction information of node 15:`
`- (15, 16, {'weight': 400.0, 'date': '2022-06-10'})`
`- (15, 17, {'weight': 200.0, 'date': '2022-07-01'})"}` |
| Assistant | **Thought:** Continue by obtaining incoming transaction information.
**Action:** `<API>graph.in_edges(15, data=True)</API>` |
| Function | `{"error": "", "response": "Direct incoming information for node 15:`
`- (10, 15, {'weight': 880.0, 'date': '2022-05-20'})"}` |
| Assistant | **Thought:** Now we've identified neighbor nodes of 15 are 10, 16, and 17. Next, analyze fund flow between these neighbors using `<API>graph.subgraph([10, 16, 17]).edges(data=True)</API>`.
**Action:** `<API>graph.subgraph([10, 16, 17]).edges(data=True)</API>` |
| Function | `{"error": "", "response": "(10, 16, {'weight': 300.0, 'date': '2022-06-05'})`
`- (10, 17, {'weight': 100.0, 'date': '2022-06-15'})`
`- (16, 17, {'weight': 50.0, 'date': '2022-07-10'})"}` |
| Assistant | **Thought:** Based on this information, we can draw conclusions: Node 10 transferred 880 yuan to node 15, 300 yuan to node 16, and 100 yuan to node 17. There is also fund flow between nodes 16 and 17, with node 16 transferring 50 yuan to node 17.
**Action:** `<API>Finish->answer</API>`
**Answer:** Node 10 not only transferred 880 yuan to node 15, but also transferred 300 yuan to node 16 and 100 yuan to node 17. There is also fund flow between nodes 16 and 17, specifically, node 16 transferred 50 yuan to node 17. |

- **Social Networks**: From the Facebook and Twitter datasets
- **Citation Graphs**: From the Cora, CiteSeer, and PubMed datasets
- **Traffic Networks**: From the METR-LA dataset

For simplicity, we simplified the graph data as shown in Figure 7. Following Wang et al. [2024c], we prepared two versions of each graph to accommodate different baselines:

- For text-instruction baselines, we restricted inputs to no more than 30 nodes and 300 edges due to context length limitations
- For tool-instruction baselines, we limited inputs to no more than 100 nodes and 1000 edges

Table 12: The prompt template for constructing the RL dataset.

| Category | Description |
|---|---|
| Dataset Name | `cash_flow_graph.gexf` |
| Dataset Type | `MultiDirected graph with weights and dates` |
| Dataset Content | The fund transfer data of a specific group. Directed edge A→B means A transferred funds to B. Graph construction:
`G = nx.MultiDiGraph(), G.add_edge(sender, receiver, weight=amount, date=transfer_date),`
where "sender" and "receiver" are the transfer participants, "amount" is the money amount, and "transfer_date" is the date. Integer type is used for nodes. |
| Task | Judge the reasonableness of thought and API names based on three dimensions:
**(1)** *API Correctness*: Whether the method exists in `networkX`, accepts the specified parameters, and matches the dataset type.
**(2)** *Thought and API Effectiveness*: How directly and effectively this step contributes to solving the user question.
**(3)** *Graph Distillation*: Whether the current thought and API can reduce information content or narrow search scope. |
| Output Type | `json` |
| Output Rules | **(1)** Output must be json data containing "apiResult" as shown in the example.
**(2)** Output can have only this json data with no additional information. |
| Special Attention | Output in "apiResult" should follow:

`{"api_Correctness": "", "thoughtAndApi_Effectiveness": "", "Graph_Distillation": ""}` |

This approach ensures fair comparison across all methods while allowing us to evaluate `GraphChain`'s scalability advantages with full-sized graphs in our main experiments.

# G  Complexity Analysis of Structure-aware Test-Time Adaptation

This section analyzes the computational complexity of our Structure-aware Test-Time Adaptation (STTA) mechanism.

## G.1  Graph Structural Fingerprinting

For a graph $G$ with $N$ nodes and $E$ edges, computing the $M$ smallest singular values of the normalized Laplacian has:

- Time complexity: $\mathcal{O}(E \cdot M \cdot T_{\text{iter}})$, where $T_{\text{iter}}$ is the number of iterations in the iterative SVD algorithm
- Space complexity: $\mathcal{O}(N + E + NM)$

We employ iterative methods (Lanczos algorithm or power iterations) instead of full SVD to efficiently compute only the needed singular values. Since $M \ll N$ (typically $M = 10$ to $50$), this computation remains efficient even for large graphs.

## G.2  Structure-Conditioned Prompt Generation

The adapter network $\mathcal{A}_\psi$ that maps the structural fingerprint to soft prompts has:

- Time complexity: $\mathcal{O}(M \cdot H + H \cdot L_p \cdot d_{emb})$
- Space complexity: $\mathcal{O}(M \cdot H + H \cdot L_p \cdot d_{emb})$

Table 13: Examples from the RL dataset with evaluation scores.

| Category | User Question | Thought and API |
|---|---|---|
| Input | Please analyze whether there is a fund circulation flow in the graph, and identify all nodes involved in the circulation flow. Additionally, please calculate the maximum amount of funds involved in these circulation paths. Begin! | We obtained the minimum cut of the maximum fund flow from node 1 to node 3 and its corresponding maximum flow value X. This means that, after considering all possible flow paths, the maximum fund transfer amount on the path from node 1 to node 3 is X. Additionally, we obtained the two sets that constitute the minimum cut, which helps us understand which nodes are involved in this maximum fund transfer process. The next step is to list the names of all nodes involved in the circulation flow to display directly to the user. `API: G.nodes()` |
| Output | | `API Correctness: 0.8, Thought and API Effectiveness: 0.9, Graph Distillation: 0.0` |
| Input | Please analyze whether there is a fund circulation flow in the graph, and identify all nodes involved in the circulation flow. Additionally, please calculate the maximum amount of funds involved in these circulation paths. Begin! | To complete this task, we need to proceed step by step. First, we need to determine whether there is a fund circulation flow in the graph, which can be achieved by calling `<API>nx.find_cycle(G)</API>` to return a list of cycle edges if a cycle exists. Next, we need to identify all nodes involved in the circulation flow. `<API>nx.simple_cycles(G)</API>` can be used to find all simple cycles (i.e., cycles that do not repeat any node) in the graph. Finally, for each identified circulation path, we need to calculate its maximum fund amount, which can be done by traversing each edge in the cycle and accumulating their weights. `API: nx.find_cycle(G)` |
| Output | | `API Correctness: 0.9, Thought and API Effectiveness: 0.95, Graph Distillation: 1.0` |

Where $H$ is the hidden dimension, $L_p$ is the prompt length, and $d_{emb}$ is the embedding dimension. This adapter is extremely lightweight (0.01%-0.1% of LLM parameters).

## G.3 Self-Supervised Adaptation

The REINFORCE-based adaptation using $K$ auxiliary queries with $R$ rollouts per query has:

- Time complexity: $\mathcal{O}(K \cdot R \cdot \bar{N} \cdot C_{LLM} + K \cdot R \cdot \bar{N} \cdot C_{KL})$
- Space complexity: $\mathcal{O}(K \cdot R \cdot \bar{N} + |\psi|)$

Where $\bar{N}$ is the average chain length, $C_{LLM}$ is the cost of an LLM forward pass, $C_{KL}$ is the cost of computing KL divergence, and $|\psi|$ is the parameter count of the adapter.

## G.4 Overall Efficiency

The total computational cost can be summarized as:

$$C_{total} = \mathcal{O}(E \cdot M \cdot T_{iter}) + \mathcal{O}(K \cdot R \cdot \bar{N} \cdot C_{LLM}) + \mathcal{O}(T_{query} \cdot \bar{N}_{query} \cdot C_{LLM}) \quad (13)$$

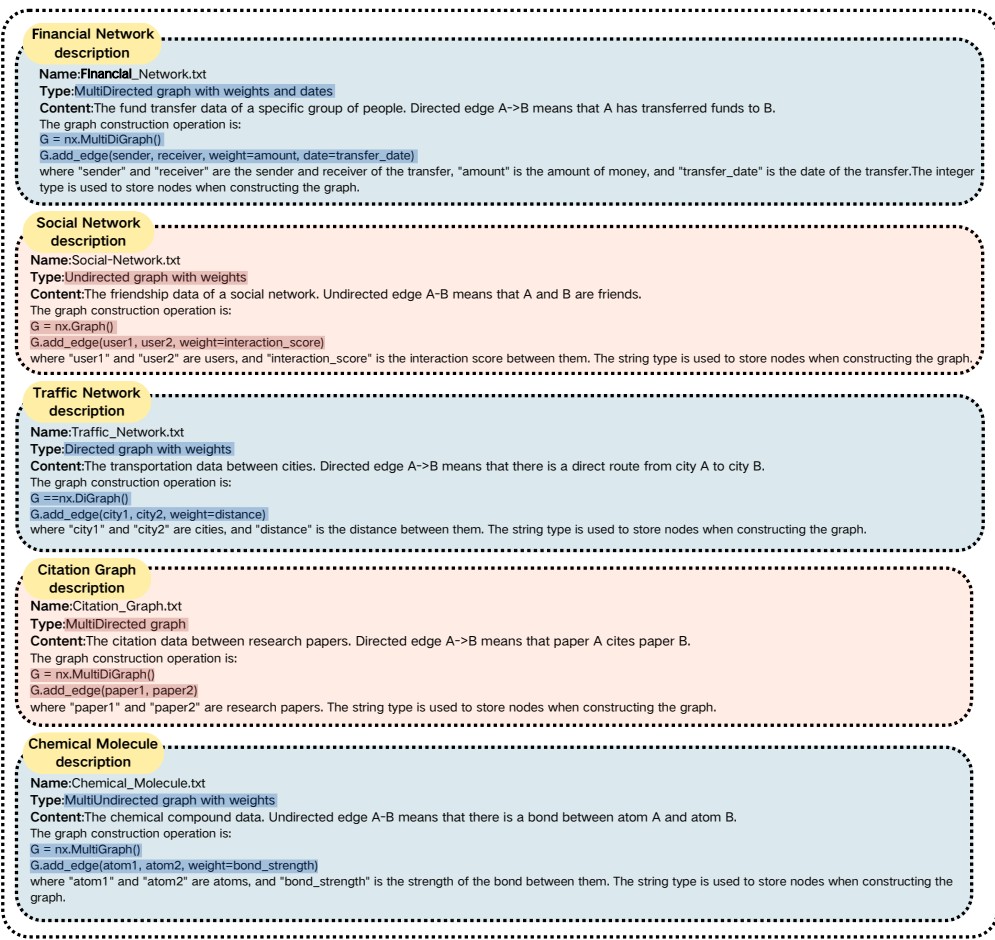

Figure 7: Detailed Description of the Graph Datasets for the Five Scenarios.

Our approach is efficient because: (1) graph fingerprinting is performed only once per graph; (2) adaptation requires few rollouts (typically $K = 5$, $R = 3$); and (3) only the small adapter network needs updating.

# H   Broader Impact

GraphChain's ability to process large-scale graphs efficiently could significantly enhance data analysis capabilities in critical domains such as financial fraud detection, healthcare networks, and social network analysis. By enabling more effective reasoning over complex interconnected data, GraphChain could help identify suspicious transaction patterns, improve epidemiological network analysis, and better understand information propagation in social networks. The framework's adaptability across diverse graph structures makes it particularly valuable for interdisciplinary research and applications where domain experts need to analyze graph data without specialized technical knowledge. Moreover, the reduced computational requirements of our approach compared to retraining models for each new graph domain could lead to more environmentally sustainable AI deployments by decreasing the energy consumption associated with large-scale model training. These advancements contribute to more accessible, efficient, and effective graph analytics tools that can address various societal challenges.

