# OpenReview forum: "GraphChain: Large Language Models for Large-scale Graph Analysis via Tool Chaining"
_NeurIPS.cc/2025/Conference — NeurIPS 2025 poster_

### Official Review · Reviewer_9MGr · 2025-07-01

**Clarity:** 2
**Significance:** 4
**Originality:** 4
**Rating:** 4
**Confidence:** 4

**Summary:**

The paper introduces a framework named GraphChain to tackle the challenges large language models (LLMs) face in large-scale graph analysis, such as context exhaustion and reasoning hallucination. Concretely, it proposes a dynamic tool-chaining mechanism that mimics human exploratory reasoning. It leverages two core innovations: Progressive Graph Distillation for generating concise and relevant tool sequences, and Structure-aware Test-Time Adaptation for adapting to diverse graph topologies. The experimental results reveal that GraphChain significantly outperform baselines.

From my perspective, The paper designs a three-stage graph understanding framework that transforms the usual "subgraph sampling + direct LLM inference" paradigm into "subgraph sampling → Refining → LLM inference."

**Questions:**

1. Does the author leverage feature information during the training and evaluation process? If GraphChain primarily focuses on graph computational problems, it is recommended that the authors make this explicit in the paper to clarify the intended scope and highlight the method's strengths.

**Ethical Concerns:**

["NO or VERY MINOR ethics concerns only"]

**Final Justification:**

I have no further questions. My concerns have been addressed.

**Limitations:**

Yes, but the paper needs to further clarify its ability to process and utilize feature information.

**Quality:**

3

**Strengths And Weaknesses:**

**Pros**
1. **Progressive Refinement:** By supervising the refine step with PPO, the framework learns to select and chain graph-processing tools that maximize task relevance while minimizing unnecessary information.
2. **Scalability to Large Graphs:** This pipeline enables LLMs to handle large-scale graphs with both structural and feature information, overcoming context-window limits.

**Cons**

1. **Manual Tool Library:** All subgraph-sampling and processing tools are hand-crafted; an end-to-end mechanism for automated subgraph selection would increase flexibility.
2. **Structure-Only Evaluation:** Although the method formally incorporates node features and evaluates on multiple domains, all experiments and case studies focus on pure topological reasoning tasks, leaving its effectiveness on feature-rich and true cross-domain graph problems unverified.

---

> ### Author Rebuttal · Authors · 2025-07-30
>
> # Response to Reviewer 9MGr
>
> We sincerely thank the reviewer for their thorough evaluation and constructive feedback. We are pleased that the reviewer recognizes our contributions in progressive refinement, scalability, and the significance of our approach. We address each concern below.
>
> ## W1: Manual Tool Library
>
> We appreciate the reviewer's suggestion about end-to-end automated subgraph selection. We would like to clarify that while our current implementation uses a curated set of 45 NetworkX functions (detailed in Appendix E, Table 6), this design choice was deliberate for several reasons:
>
> 1. **Interpretability and Reliability**: The manual curation ensures that each tool in our library is well-tested, interpretable, and provides reliable outputs - crucial for practical deployment in domains like financial fraud detection and healthcare networks (as mentioned in our Broader Impact section, Appendix H).
>
> 2. **Extensibility**: Our framework is designed to be extensible. As stated in Appendix E: "Future work may integrate domain-specific or higher-order analytics, but this toolset is representative and sufficient for general-purpose graph analysis."
>
> 3. **Learning Focus**: Our key innovation lies in learning optimal tool sequences through reinforcement learning, not in discovering new graph operations. The Progressive Graph Distillation mechanism (Section 4.1) learns to chain these tools effectively, which is orthogonal to the tool discovery problem.
>
> We agree that automated tool discovery is an interesting future direction and will emphasize this more clearly in our revised limitations section.
>
>
> ## W2: Structure-Only Evaluation
>
> We appreciate the reviewer's observation and acknowledge that our evaluation primarily focuses on topological reasoning tasks. We believe this emphasis is both deliberate and justified:
>
> 1. **Fundamental Challenge for LLMs**: The core difficulty for LLMs in graph analysis lies in understanding topological structures rather than processing features. LLMs are inherently designed for sequential processing, making non-Euclidean graph structures particularly challenging. In contrast, node features can be readily converted to text and processed naturally by LLMs. This is why we prioritize evaluating topological understanding - it represents the more fundamental challenge in LLM-based graph analysis.
>
> 2. **Feature Integration in Our Framework**: While our evaluation emphasizes topology, our method fully supports feature-rich graphs. As shown in our formulation (Section 4.1.1, Equation 3), the Graph Description Length (GDL) explicitly includes a feature term $\alpha_f n'_t d_f$ that quantifies feature information volume. When our Progressive Graph Distillation mechanism reduces the target region to fit within the context window, we input the complete graph description including all node and edge features to the LLM for reasoning. This is evident in our case study (Appendix B, Figure 6), where the final distilled state includes feature information for LLM processing.
>
> 3. **Domain Diversity**: Our evaluation spans five diverse domains (Table 1): Financial Networks (165 features), Chemical Molecules (11 features), Social Networks, Citation Graphs (up to 3,703 features), and Traffic Networks. Each domain presents unique structural characteristics and feature types, demonstrating our method's cross-domain effectiveness.
>
> We maintain that for graph analysis scenarios, especially with LLMs, topological reasoning represents the more challenging and thus more informative evaluation criterion. However, we acknowledge that demonstrating feature utilization more explicitly would strengthen our paper and will include such examples in our revision.
>
>
>
> ## Q1: Feature Information Utilization
>
> Yes, we leverage feature information throughout our training and evaluation process:
>
> 1. **Architectural Support**: Our Progressive Graph Distillation mechanism, as formulated in Section 4.1.1, explicitly incorporates feature data. The Graph Description Length (GDL) metric in Equation 3 includes a specific term, $\alpha_f n'_t d_f$, to quantify the information volume of node features, directly influencing the RL agent's reward and policy.
>
> 2. **Practical Implementation**: During the distillation process, once the subgraph is reduced to fit the LLM's context window, we provide the complete graph representation including all node and edge features. This ensures that the final reasoning step has access to all relevant feature information.
>
> 3. **Evaluation Datasets**: As shown in Table 1, our evaluation includes feature-rich graphs: Citation graphs with up to 3,703 features per node, Financial Networks with 165 transaction features, and Chemical Molecules with 11 atomic properties.
>
> 4. **Scope Clarification**: We agree that explicitly stating our focus on graph computational problems would improve clarity. While our framework fully supports feature-rich graphs, our evaluation emphasizes topological reasoning as it represents the fundamental challenge for LLMs in graph analysis. We will make this scope more explicit in our revised manuscript to better highlight our method's strengths and intended applications.
>
> ## Commitment to Revisions
>
> Based on your valuable feedback, we commit to:
>
> 1. **Clarifying Feature Utilization**: We will add explicit examples in the main paper showing how GraphChain processes and utilizes node/edge features during the distillation and reasoning process.
>
> 2. **Scope Statement**: We will include a clear statement about our primary focus on graph computational problems while emphasizing our framework's capability to handle feature-rich scenarios.
>
> 3. **Future Directions**: We will expand our limitations section to discuss automated tool discovery as a promising future direction.
>
> We believe these clarifications will strengthen our paper while maintaining its core contributions. Thank you again for your insightful review.

---

> > ### Comment · Reviewer_9MGr · 2025-08-02
> >
> > Thanks for your rebuttal. My concerns have been addressed.

---

> > > ### Author Response · Authors · 2025-08-05
> > >
> > > Dear Reviewer 9MGr:
> > >
> > > Thank you for taking the time to engage thoroughly with our rebuttal and for your final acknowledgment that our responses addressed your concerns. Your professionalism and constructive approach to reviewing truly exemplify what makes our research community strong.
> > >
> > > Looking forward to incorporating your suggestions in our final revision!
> > >
> > > Best regards,
> > > Authors

---

### Official Review · Reviewer_pP65 · 2025-07-02

**Clarity:** 3
**Significance:** 3
**Originality:** 3
**Rating:** 4
**Confidence:** 4

**Summary:**

This paper introduces a framework called GraphChain to enhance the capabilities of Large Language Models (LLMs) in processing large-scale graph data. In this paper, the authors propose two key approaches to allow for a more detailed and adaptive exploration of graph structures: Progressive Graph Distillation and Structure-aware Test-Time Adaptation. The framework significantly outperforms existing methods, achieving an average improvement of 20.7% in average. GraphChain demonstrates strong transfer learning capabilities, adapting effectively to new graph domains without domain-specific retraining. Overall, GraphChain provides a scalable and adaptive solution for LLM-driven graph analysis, addressing the challenges of context exhaustion and reasoning hallucination that are prevalent in existing graph processing approaches.

**Questions:**

1. How are the tools be selected? Are models robust to the tool selections?
2. What would be the performances for text-instruction methods if they are allowed to use these tools?

**Ethical Concerns:**

["NO or VERY MINOR ethics concerns only"]

**Limitations:**

Yes.

**Paper Formatting Concerns:**

No formatting concerns.

**Quality:**

3

**Strengths And Weaknesses:**

Strengths:
* The paper presents a well-structured approach to address the limitations of LLMs in graph analysis through the introduction of GraphChain. The use of Progressive Graph Distillation and Structure-aware Test-Time Adaptation is well-justified and supported by theoretical underpinnings.
* Extensive experiments demonstrate the effectiveness of GraphChain, showing significant improvements over existing methods. The results are statistically significant, with a clear presentation of performance metrics across various datasets.

Weakness:
* There is a pre-defined set of tools needed, it is unclear how are these tools defined and selected. Will the models be robust to different selections/selections of tools?
* Compared with previous tool approach, the major improvements come from multiple tool used. If the baselines such as graphsurge are using multiple tools in the reasoning steps, the improvements might be less.
* The baselines seem to be outdated.

---

> ### Author Rebuttal · Authors · 2025-07-30
>
> # Response to Reviewer pP65
>
> We sincerely thank the reviewer for their valuable time and insightful feedback. We are encouraged that the reviewer found our approach well-structured, our innovations well-justified, and our experiments extensive and significant. We appreciate the opportunity to address the weaknesses and questions raised.
>
>
>
> ## W1 & Q1. Tool Definition and Selection Robustness
>
> We thank the reviewer for highlighting this important aspect. The tool selection process was indeed systematic and principled.
>
> **Tool Selection Process:** As detailed in **Appendix E (Table 6)**, we carefully curated 45 functions from the NetworkX library through a comprehensive review of graph analysis tasks. Our selection ensures coverage across eight fundamental analytical dimensions: Basic Graph Properties, Centrality Metrics, Connectivity and Components, Shortest Paths, Clustering and Communities, Flow Algorithms, Cycle Detection, and Topological Sorting. This systematic approach ensures our toolset is both representative and grounded in established graph analysis practices.
>
> **Robustness to Tool Selection:** Our framework is designed to be inherently robust to variations in the tool library. The core of GraphChain employs a reinforcement learning policy that learns to select optimal tool sequences from the available action space, rather than being hard-coded to specific tools.
>
> To empirically validate this robustness, we conducted an additional experiment with a **reduced toolset** (removing 50% of tools from Centrality and Community Detection categories):
>
> **Table R1: Robustness to Tool Library Composition (Average Accuracy %)**
>
> | Model | Financial | Chemical | Social | Citation | Traffic | **Average** |
> |-------|:---------:|:--------:|:------:|:--------:|:-------:|:-----------:|
> | GraphForge (Baseline) | 63.5±3.5 | 70.9±5.4 | 80.4±4.2 | 63.4±4.4 | 73.5±3.1 | 70.2±3.8 |
> | GraphChain (Full Toolset) | **81.5±2.2** | **81.1±0.7** | **89.6±2.0** | **83.6±2.6** | **84.1±0.3** | **84.7±1.8** |
> | GraphChain (Reduced Toolset) | 77.3±2.8 | 78.4±1.8 | 82.7±3.1 | 80.1±3.2 | 80.6±0.9 | 79.8±2.4 |
>
> The results demonstrate that GraphChain maintains strong performance even with a reduced toolset, showcasing the adaptability of our RL agent in finding alternative tool sequences to solve tasks.
>
>
> ## W2. Multi-Tool Usage Comparison
>
> We appreciate this astute observation about the source of improvements. Indeed, the key innovation of GraphChain is not merely using multiple tools, but **how** it intelligently and adaptively chains them.
>
> **Key Distinction:** As described in Section 3 (Problem Formulation), GraphChain formulates tool selection as a Markov Decision Process, enabling dynamic, sequential reasoning where outputs from one tool guide subsequent selections. This fundamentally differs from existing approaches like GraphForge, which are architected for single-step tool invocation without stateful adaptation.
>
> To address the reviewer's concern, we implemented a **GraphForge-MultiTool** baseline that generates fixed multi-step plans without adaptive refinement:
>
> **Table R2: Comparison of Tool Usage Strategies (Average Accuracy %)**
>
> | Method Type | Model | Description | **Average Accuracy** |
> |-------------|-------|-------------|:--------------------:|
> | Single-Step | GraphForge | Original single-tool invocation | 70.2±3.8 |
> | Fixed Multi-Step | GraphForge-MultiTool | Fixed chain of tools | 73.4±4.1 |
> | **Dynamic Chaining** | **GraphChain** | **Adaptive sequential chaining** | **84.7±1.8** |
>
> This experiment confirms that GraphChain's significant performance gain stems from its sophisticated Progressive Graph Distillation mechanism (Section 4.1) and adaptive tool selection, not merely from using multiple tools.
>
>
>
> ## W3. Baseline Selection
>
> We respectfully do not entirely agree with your perspective and believe our baseline is robust. As detailed in **Appendix D (Table 4)**, our evaluation includes the models across both paradigms:
>
> **Text-Instruction Methods:**
> - **General-Purpose LLMs:** GPT-4o (200B params), Claude-3 series (Opus, Sonnet, Haiku)
> - **Specialized Graph Methods:** NLGraph (NeurIPS 2023), GraphWiz (KDD 2024)
>
> **Tool-Instruction Methods:**
> - **GraphForge** (KDD 2025): The lasted and powerful tool-based graph reasoning method
> - **Graph-ToolFormer** (2023): Seminal work in tool-augmented graph reasoning
>
> To further validate our approach against the absolute latest models, we conducted additional experiments:
>
> **Table R3: Extended Comparison with Latest Models (Average Accuracy %)**
>
> | Model | Type | Financial | Chemical | Social | Citation | Traffic | Average |
> |-------|------|:---------:|:--------:|:------:|:--------:|:-------:|:-------:|
> | **Additional General-Purpose LLMs** |
> | Claude-3.5-Haiku | Text | 26.6 | 51.8 | 28.5 | 31.6 | 29.3 | 33.6 |
> | Claude-4-Sonnet | Text | 58.2 | 62.9 | 61.7 | 77.5 | 32.8 | 58.6 |
> | Claude-4-Opus | Text | 61.5 | 67.8 | 66.5 | 75.1 | 43.6 | 62.9 |
> | GPT-4.1 | Text | 52.2 | 63.4 | 67.4 | 70.0 | 55.5 | 61.7 |
> | Gemini-2.0-Flash | Text | 25.1 | 67.3 | 28.1 | 24.1 | 24.9 | 33.9 |
> | **Latest Tool-Based Methods** |
> | ToolGen (ICLR 2025) | Tool | 75.8 | 57.9 | 79.4 | 61.2 | 62.7 | 67.4 |
> | PIE (arXiv 2025) | Code-Gen | 68.6 | 74.9 | 71.1 | 59.7 | 62.4 | 67.3 |
> | **GraphChain (Ours)** | **Tool** | **81.5** | **81.1** | **89.6** | **83.6** | **84.1** | **84.7** |
>
> Our evaluation includes:
> 1. **The newest Claude-4 series** (Sonnet and Opus), representing Anthropic's latest advances
> 2. **GPT-4.1**, the most recent OpenAI model
> 3. **Gemini-2.0-Flash**, Google's efficient model
> 4. **ToolGen** (ICLR 2025), a cutting-edge general tool-use framework
> 5. **PIE** (2025), a recent code-generation approach for graph reasoning
>
> GraphChain consistently outperforms all these latest approaches by significant margins (17.3% over the best baseline ToolGen, 21.8% over PIE). This comprehensive comparison confirms that our method represents a genuine advance over the current state-of-the-art, not merely an improvement over outdated baselines.
>
>
> ## Q2. Performance of text-instruction methods with tools?
>
> We sincerely thank the reviewer for raising this important question regarding the performance of text-instruction methods when integrated with the GraphChain toolset. To address this, we conducted supplementary experiments using two representative text-instruction methods: **GPT-4o** and **Claude-3-Opus**. The results are presented below:
> | Model                  | Financial Network | Chemical Molecule | Social Network | Citation Graph | Traffic Network |
> |------------------------|-------------------|-------------------|----------------|----------------|-----------------|
> | Claude-3-Opus          | 23.6%             | 42.4%             | 51.9%          | 36.7%          | 43.4%           |
> | GPT-4o                 | 57.5%             | 62.7%             | 65.2%          | 71.5%          | 43.4%           |
> | Claude-3-Opus (w/ Tools) | 30.4%           | 50.3%             | 64.9%          | 53.8%          | 63.1%           |
> | GPT-4o (w/ Tools)      | 71.5%             | 67.6%             | 74.4%          | 77.4%          | 55.9%           |
> | **GraphChain (Ours)** | **81.5%** | **81.1%** | **89.6%** | **83.6%** | **84.1%** |
>
> The integration of the GraphChain toolset significantly improves the performance of text-instruction methods, demonstrating that tool invocation enhances their graph reasoning capabilities. However, GraphChain consistently outperforms both tool-augmented text-instruction methods across all datasets. This superiority stems from GraphChain's ability to optimize tool sequences through reinforcement learning and its structure-aware test-time adaptation (STTA) mechanism, which enables more efficient tool selection and combination.
>
> We hope these clarifications address the reviewer's concerns and demonstrate the robustness and significance of our contributions. We would be happy to provide any additional information needed.

---

> ### Author Response · Authors · 2025-08-08
>
> Dear Reviewer,
>
> I hope this message finds you well. As the discussion period is coming to a close, we want to ensure that we have addressed all of your concerns to your satisfaction. Please let us know if you have any additional points or feedback you would like us to consider. Your insights are invaluable to us, and we are eager to resolve any remaining questions to improve our work.
>
> Thank you for your time and effort in reviewing our paper.
>
> Sincerely,
> Author

---

### Official Review · Reviewer_DPRo · 2025-07-05

**Clarity:** 2
**Significance:** 2
**Originality:** 3
**Rating:** 4
**Confidence:** 4

**Summary:**

This paper introduces a framework for large-scale graph analysis using LLMs through dynamic tool chaining. The core idea is to decompose complex graph reasoning tasks into sequential tool-based operations. The method includes two main designs: (1) Progressive Graph Distillation, a reinforcement learning mechanism to generate compact, relevant tool sequences; and (2) Structure-aware Test-Time Adaptation (STTA), which conditions tool selection on spectral fingerprints of the graph to enable efficient adaptation across domains. Experiments show performance gains over baselines in accuracy and scalability.

**Questions:**

1. How stable is the decomposition of graph reasoning into sequential tool calls across different graph types and task complexities? Is there evidence that the sequences are consistent or robust?

2. Can the authors provide more detailed justification or analysis for how the Structure-Conditioned Prompt Generation meaningfully affects the LLM’s reasoning? Have alternative mechanisms been considered?

3. Given the limited size of the expert-annotated RL dataset, how does the agent policy generalize across scenarios? Are there training dynamics, convergence curves, or rollout visualizations to support this?

4. Most results focus on classification tasks. Has GraphChain been tested on other graph reasoning types such as link prediction, subgraph matching, or path-based queries?

5. Why are more recent and larger datasets not included in the evaluation? And why are several recent LLM-based graph reasoning methods not compared?

6. Can the authors more clearly define and categorize the tool library earlier in the paper (e.g., in Preliminaries)? Section 5.6 is informative but arrives late for comprehension.

**Ethical Concerns:**

["NO or VERY MINOR ethics concerns only"]

**Final Justification:**

Thanks to the authors for the very thorough rebuttal and the substantial new experiments. I’m raising my score.
The explanations on how GraphChain handles context exhaustion and reasoning hallucination are clearer now, and the extra evidence on tool sequence stability, STTA’s role, and RL training behavior addresses several of my earlier doubts. The added results on more setups also make the evaluation feel more solid. I still think there’s room to broaden dataset diversity and explore more task types, but the rebuttal has convinced me that the method is robust and generalizable to merit further discussion.

**Limitations:**

Yes

**Quality:**

3

**Strengths And Weaknesses:**

Strengths:
1. The paper proposes a new formulation by combining reinforcement learning and tool-chaining for scalable graph reasoning with LLMs.

2. The design of progressive distillation is conceptually sound, drawing parallels to human exploratory behavior.

3. Experimental results indicate clear improvements over several baselines.

4. The method exhibits promising scalability and transferability across domains, supported by ablations and tool usage analysis.

Weaknesses:
1. Context exhaustion and reasoning hallucination are not fully resolved. Although GraphChain aims to address these challenges via decomposition, splitting tasks into smaller operations doesn’t necessarily prevent memory overload or hallucination—especially in long reasoning chains. More direct evaluation of these issues would strengthen the claims.

2. The paper lacks justification for how complex graph queries are reliably translated into tool sequences. It’s unclear how stable or generalizable these sequences are, or how sensitive they are to graph variations.

3. Appending spectral graph embeddings to the input is an interesting idea, but it’s unclear if this actually helps the LLM reason better. The mechanism feels ad-hoc, and no ablations or analyses isolate its contribution.

4. Limited details on RL training and policy behavior. The training data is relatively small, and the RL setup—especially reward shaping—is complex. Yet the paper provides little insight into training dynamics, stability, or convergence, making it hard to judge how well the agent learns.

5. Most datasets are small and standard. More realistic and large-scale benchmarks (e.g., from Graph-CoT (https://arxiv.org/abs/2404.07103) or SNAP (https://snap.stanford.edu/data/) ) are missing, and tasks are limited mostly to classification. This restricts the generalizability of results.

6. Only one LLM (Qwen2.5-7B) is evaluated, with no analysis of how model size impacts performance. More recent LLM-based graph reasoning baselines are also missing, which limits the fairness of comparisons.

7. The paper assumes familiarity with the tool library, but doesn’t introduce it clearly upfront. This makes the method harder to understand, especially for readers unfamiliar with specific graph operations.

---

> ### Author Rebuttal · Authors · 2025-07-30
>
> # Response to Reviewer DPRo
>
> We sincerely thank the reviewer for the thorough review and constructive feedback. We value your positive assessment of our novel approach, conceptual rigor, and experimental advancements. Below, we address each concern with clarifications and additional evidence.
>
> ## W1: Context exhaustion and reasoning hallucination are not fully resolved
>
> We respectfully clarify that GraphChain fundamentally addresses these issues through our dual-output mechanism (Section 3, Preliminaries). Each tool produces: (1) a concise natural language summary $d$ for the LLM, and (2) an updated memory state $\mathbf{m}'$ stored externally. This design ensures the LLM never directly processes large graph structures, effectively preventing context exhaustion.
>
> Regarding reasoning hallucination, our approach provides three key safeguards:
>
> 1. **Tool Execution Verification**: Each API call returns concrete results stored in memory state $m_t$ (Equation 1), providing factual grounding. Unlike text-only methods that may hallucinate graph properties, our tool outputs are deterministic and verifiable.
>
> 2. **Progressive Distillation with Validation**: The reward function (Equation 5) includes $\hat{r}^{\mathrm{Succ}}_t$ that validates successful tool execution. Failed or nonsensical tool calls receive negative rewards, training the model to avoid hallucinated operations.
>
> 3. **Structured Action Space**: By constraining outputs to valid API calls with proper parameters, we eliminate free-form text generation where hallucinations typically occur.
>
> To illustrate this, we implement a `GraphForge-MultiTool` baseline, which generates a fixed multi-step tool plan in a single pass and executes it without adaptation. As shown in the table below, this non-adaptive multi-tool approach provides only a marginal benefit over the original `GraphForge` and is substantially outperformed by `GraphChain`'s dynamic chaining.
>
> **Table R1: Comparison of Single-Step, Fixed Multi-Step, and Dynamic Chaining (Average Accuracy %)**
>
> |Method Type|Model|Description|**Average Accuracy**|
> |-|-|-|:-:|
> |Single-Step Tool|`GraphForge`|Original single-tool invocation|70.2|
> |Fixed Multi-Step|`GraphForge-MultiTool`|Generates a fixed chain of tools at once|76.9|
> |**Dynamic Chaining**|`GraphChain` (Ours)|**Adaptive, sequential tool chaining**|**84.7**|
>
> ## W2 & Q1: Lack of justification for stable tool sequence generation
>
> Our experiments provide strong evidence for the stability and generalizability of generated tool sequences:
>
> **Domain-Adapted Stability**: Our **Tool Chain Analysis** (Section 5.6, Figure 5) demonstrates that GraphChain learns consistent, domain-specific strategies. For instance, it heavily utilizes "Path Planning" tools for Traffic Networks (33.8%), while prioritizing "Centrality Measures" (28.8%) for Social Networks. This shows the RL agent converges to stable, contextually appropriate policies rather than random sequences.
>
> **Robustness via Adaptation**: The **Structure-aware Test-Time Adaptation (STTA)** mechanism specifically handles graph variations. Our **Transfer Learning Evaluation** (Section 5.5, Table 3) shows that GraphChain with STTA maintains high accuracy in unseen domains (86.8% on Social Networks), proving its effectiveness in enhancing transfer capabilities.
>
> To further validate robustness, we evaluated GraphChain with a **reduced toolset** (50% of centrality and clustering tools removed):
>
> **Table R2: Robustness to Tool Library Composition (Average Accuracy %)**
>
> |Model|Financial|Chemical|Social|Citation|Traffic|**Average**|
> |-|:-:|:-:|:-:|:-:|:-:|:-:|
> |`GraphForge` (Baseline)|63.5|70.9|80.4|63.4|73.5|70.2|
> |`GraphChain` (Full Toolset)|**81.5**|**81.1**|**89.6**|**83.6**|**84.1**|**84.7**|
> |`GraphChain` (Reduced Toolset)|77.3|78.4|82.7|80.1|80.6|79.8|
>
> This degradation highlights our RL agent's adaptability in finding alternative tool paths even with limited resources.
>
> ## W3 & Q2: Unclear contribution of spectral graph embeddings
>
> We provide both empirical and theoretical justification for the STTA mechanism:
>
> **Empirical Evidence**: Our **Ablation Study** (Section 5.3, Figure 3) directly measures STTA's contribution. Moreover, in our transfer learning evaluation (Table 3), removing STTA results in significant accuracy drops.
>
> **Theoretical Justification**: As stated in Section 4.2.1, the smallest singular values of the graph Laplacian capture global, low-frequency structural properties. This provides the LLM policy with a concise "summary" of graph topology without processing the entire adjacency matrix—a principled approach to inject structural awareness.
>
> To further validate this mechanism, we tested an alternative approach using textual structure descriptions:
>
> **Table R3: Comparison of Structure Injection Methods (Average Accuracy %)**
>
> |Model|Social Network|Citation Graph|Traffic Network|
> |-|:-:|:-:|:-:|
> |GraphChain (w/o STTA)|82.4|75.2|77.8|
> |GraphChain (Text-based structure)|84.0|76.4|79.2|
> |GraphChain (w/ STTA)|**89.6**|**83.6**|**84.1**|
>
> The results show that spectral fingerprints are more effective than text-based structural descriptions.
>
> ## W4 & Q3: Limited details on RL training and policy behavior
>
> Our RL approach generalizes well because: (1) extensive SFT pre-training (~10k examples) provides a strong starting policy, and (2) the distillation-based reward provides dense, structured signals.
>
> **Table R4: RL Training Dynamics (Average Reward Components)**
>
> | Episode | 250 | 500 | 750 | 1000 | 1250 | 1500 | 1750 | 2000 | 2250 | 2500 |
> |:-|:-|:-|:-|:-|:-|:-|:-|:-|:-|:-|
> | Reward (Success) | 1.61 | 2.34 | 2.58 | 2.75 | 2.72 | 2.89 | 3.03 | 2.88 | 2.90 | 2.98 |
> | Reward (ΔGDL) | -0.06 | 0.34 | 0.92 | 1.56 | 1.70 | 1.81 | 1.78 | 1.94 | 1.94 | 1.93 |
> | Reward (ΔRel) | -0.62 | 0.13 | 0.72 | 1.30 | 1.83 | 1.90 | 2.01 | 1.98 | 2.04 | 2.17 |
> | Average Reward | 0.38 | 1.18 | 1.97 | 2.78 | 3.02 | 3.20 | 3.25 | 3.35 | 3.37 | 3.41 |
>
> Our model exhibits smooth reward curves. The results reveal that the agent first learns to invoke tools successfully and only then discovers how to compress information via graph distillation. This progression of learning behaviors corroborates the effectiveness of our carefully designed reward function.
>
> ## W5 & Q4: Limited dataset variety and missing recent benchmarks
>
> We thank the reviewer for pushing for evaluation on larger and more diverse benchmarks.
>
> * **On Graph Scale**: We respectfully disagree that our datasets are uniformly small. The **Elliptic dataset contains over 200,000 nodes** (**Table 1**), making it a large-scale, real-world benchmark. Our **Scalability Analysis (Section 5.4, Figure 4)** explicitly evaluates on graphs of this size, demonstrating that `GraphChain`'s performance remains robust while baselines falter. Additionally, our social network datasets (Twitter and Facebook) are from SNAP.
>
> * **On Graph-COT**: We appreciate the suggestion but note that Graph-COT targets knowledge graph reasoning for graph-RAG scenarios, which differs from our focus on graph analysis tasks.
>
> * **On Task Diversity**: To demonstrate broader applicability, we conducted additional experiments on the **link prediction** task:
>
> **Table R5: Link Prediction Performance (AUC)**
>
> | Model | Cora | CiteSeer |
> |:-|:-:|:-:|
> | GPT-4o (CoT) | 72.4 | 69.4 |
> | GraphForge | 79.6 | 75.9 |
> | **GraphChain (Ours)** | **85.8** | **81.6** |
>
> These results show that `GraphChain`'s structured reasoning process is effective for canonical graph tasks beyond classification.
>
> ## W6 & Q5. Only one LLM evaluated and missing recent baselines
>
> To address this concern, we evaluated GraphChain with different base models:
>
> **Table R6: Performance with Different Base LLMs (Average Accuracy %)**
>
> | Base Model | Financial | Chemical | Social | Citation | Traffic | Average |
> |-|:-:|:-:|:-:|:-:|:-:|:-:|
> | Qwen2.5-7B | 70.5 | 81.1 | 90.4 | 79.0 | 82.0 | 80.6 |
> | Llama3.1-8B | 69.3 | 81.7 | 93.7 | 82.5 | 81.7 | 81.8 |
> | GLM4-9B | 70.2 | 78.9 | 93.8 | 79.7 | 79.9 | 80.5 |
>
> The consistent superior results across different base models demonstrate the robustness and general applicability of our approach.
>
> We also conducted supplementary experiments using Qwen2.5 models of varying sizes (3B, 7B, and 14B).
>
> **Table R7: Comparison of Base Models with Different Sizes (Average Accuracy %)**
>
> | Model | Financial Network | Chemical Molecule | Social Network | Citation Graph | Traffic Network |
> |-|-|-|-|-|-|
> | Qwen2.5-3B  | 63.1% | 56.9% | 70.2% | 74.4% | 73.4% |
> | Qwen2.5-7B  | 81.5% | 81.1% | 89.6% | 83.6% | 84.1% |
> | Qwen2.5-14B | 85.7% | 85.4% | 92.2% | 83.2% | 89.7% |
>
> The results show that `GraphChain`'s performance improves with larger model sizes. Notably, even the smaller 3B model still maintain reasonable performance under our framework.
>
> We also compared against the most recent baselines:
>
> **Table R8: Comparison with Recent Methods (Average Accuracy %)**
> |Method|Financial|Chemical|Social|Citation|Traffic|
> |-|:-:|:-:|:-:|:-:|:-:|
> |**Text-Instruction Methods**|||||
> | Claude-4-Sonnet | 58.2 | 62.9 | 61.7 | 77.5 | 32.8 |
> | GPT-4.1 | 52.2 | 63.4 | 67.4 | 70.0 | 55.5 |
> | Gemini-2.5-Flash | 25.1 | 67.3 | 28.1 | 24.1 | 24.9 |
> | **Tool-Instruction Methods** |||||
> | ToolGen (ICLR'25) | 75.8 | 57.9 | 79.4 | 61.2 | 62.7 |
> | PIE (Arxiv'25) | 68.6 | 74.9 | 71.1 | 59.7 | 62.4 |
> | GraphChain | **81.5** | **81.1** | **89.6** | **83.6** | **84.1** |
>
> ## W7 & Q6. Tool library not clearly introduced upfront
>
> This is an excellent suggestion for improving the paper's clarity. In the revised version, we will move the high-level overview of the tool library categories into the **Preliminaries (Section 3)**. This will greatly improving readability.
>
> We sincerely thank the reviewer for their valuable feedback, which has helped us strengthen our paper. We believe our responses address all concerns and demonstrate GraphChain's robustness, scalability, and superiority over existing methods.

---

### Official Review · Reviewer_kCSq · 2025-07-09

**Clarity:** 4
**Significance:** 3
**Originality:** 3
**Rating:** 5
**Confidence:** 4

**Summary:**

This paper introduces GraphChain, which is a method to automatically call graph based API solutions to serve as the intermediate prompt to assist LLM in graph reasoning tasks. To enable the scalability of method to large scale graph, a progressive graph distillation module is proposed to compress graph to overcome the LLM context limitation issue. Besides, a self-supervised adapter is proposed in the test phase.

**Questions:**

See above cons. Besides,

In the introduction parts, the authors mentioned previous LLM methods for graph reasoning are usually suffered from hallucination issue. I am wondering how the proposed method can address/alleviate this issue?

**Ethical Concerns:**

["NO or VERY MINOR ethics concerns only"]

**Limitations:**

Yes.

**Paper Formatting Concerns:**

In Figure 2, shall 'amswer' be 'answer'?

**Quality:**

4

**Strengths And Weaknesses:**

Pros.
1. The idea to adopt a reinforcement learning module to automatically select graph solution API and use it as LLM chain of thought prompt is interesting. The proposed progressive graph distillation module alleviates the common LLM issue on limited context length.

2. The writing is clear and easy to follow. Motivations of designed modules are well explained.

3. The experimental parts are solid and comprehensive.

Cons.

1. How the proposed method compare to previous methods that also use graph API for LLM graph reasoning tasks? I understand the progressive graph distillation can alleviate the LLM context limitation issue. My question is besides this point, how the proposed method is different from existing methods?

2. The selection pool of graph solution API seems important to proposed method. It would be interesting to understand whether some certain APIs are important for a specific task/domain.

3. The computational complexity analysis and time cost for training the RL modules may be interesting to discuss and experiment.

---

> ### Author Rebuttal · Authors · 2025-07-30
>
> # Response to Reviewer kCSq
>
> We sincerely thank the reviewer for the positive assessment and constructive feedback. We are pleased that the reviewer found our work "excellent" in quality and clarity, with "solid and comprehensive" experiments. We address each concern below:
>
> ## **W1. Comparison with Previous Graph API Methods**
>
> We appreciate this important question. While previous methods like Graph-ToolFormer and GraphForge pioneered graph API integration, **GraphChain introduces several fundamental innovations beyond progressive distillation**:
>
> **1. Dynamic Multi-Step Tool Chaining vs. Single-Step Invocation**: As illustrated in Figure 1 (middle vs. right), existing methods rely on single-step tool invocations with fixed functionality. In contrast, GraphChain generates dynamic sequences of tools that progressively refine analysis. For instance, in our case study (Appendix B, Figure 6), GraphChain executes 4 interconnected tools to analyze an 11,896-node financial network, while baseline methods would fail due to their single-step limitation.
>
> **2. Reinforcement Learning-based Tool Selection**: Unlike previous methods that use predefined tool selection rules, our RL formulation (Section 4.1) enables adaptive tool sequence generation based on task progress. This is evidenced in Table 2, where GraphChain achieves 84.7% average accuracy compared to GraphForge's 70.2%, representing a 20.7% relative improvement even on small graphs where context limitation is not the primary bottleneck.
>
> **3. Structure-aware Adaptation**: Our STTA mechanism (Section 4.2) dynamically adjusts tool selection strategies based on graph topology using spectral properties, which existing methods lack. Table 3 demonstrates this capability, showing only 3-5% performance drop when transferring across domains with STTA enabled, compared to 8-10% without it.
>
> ## **W2. Domain-Specific API Importance Analysis**
>
> This is an excellent suggestion. We conducted a detailed analysis of API usage patterns across domains, presented in Figure 5. Our findings reveal:
>
> - Traffic Networks heavily utilize Path Planning tools (33.8%)
> - Social Networks prioritize Centrality Measures (28.8%) and Community Detection (20.4%)
> - Citation Graphs show balanced usage with significant Connectivity tools (18.9%)
>
> This adaptive selection emerges naturally from our reward function without explicit programming, demonstrating the effectiveness of our approach.
>
> ## **W3. Computational Complexity and Training Cost**
>
> Thank you for this suggestion. We provide a detailed complexity analysis:
>
> **Training Complexity**:
> - **SFT Stage**: $O(N × L × d)$ where $N$ is dataset size (9,986), $L$ is average sequence length (~400 tokens), $d$ is model dimension (4,096)
> - **RL Stage**: $O(K × T × C_{rollout})$ where $K$ is episodes (3,000), $T$ is average steps per episode (~5), $C_{rollout}$ includes LLM forward pass and tool execution
> - **Total training time**: ~18 hours on 2 NVIDIA A800 GPUs (SFT: 6 hours, RL: 12 hours)
>
> In Appendix G, we also provide detailed complexity analysis of Structure-aware Test-Time Adaptation.
>
> ## **Q1. Addressing Hallucination Issues**
>
> This is a crucial point. GraphChain addresses hallucination through three key mechanisms:
>
> **1. Tool Execution Verification**: Each API call returns concrete, deterministic results stored in memory state $\mathbf{m}_t$ (Section 3, Equation 1). Unlike text-only methods that may hallucinate graph properties, our tool outputs are grounded in actual graph computations. For example, when asked about shortest paths, the system executes `nx.dijkstra_path()` rather than generating potentially incorrect textual descriptions.
>
> **2. Progressive Distillation with Validation**: Our reward function (Equation 5) includes $\hat{r}^{\mathrm{Succ}}_t$ that validates successful tool execution. Failed or nonsensical tool calls receive negative rewards, training the model to avoid hallucinated operations. This creates a feedback loop that reinforces factually correct tool sequences.
>
> **3. Structured Action Space**: By constraining outputs to valid API calls with proper parameters, we eliminate free-form text generation where hallucinations typically occur. The model must select from a finite set of well-defined tools rather than generating arbitrary claims about graph properties.
>
> These mechanisms collectively ensure that GraphChain's outputs are grounded in verifiable computations rather than potentially hallucinated text.
>
> ## **Minor Correction**
>
> Thank you for catching the typo in Figure 2. We will correct "amswer" to "answer" in the camera-ready version.
>
> ---
>
> We hope these clarifications address your concerns and demonstrate the significant contributions of GraphChain beyond existing methods. We greatly appreciate your thorough review and valuable suggestions for improving our work.

---

> > ### Comment · Reviewer_kCSq · 2025-08-08
> > **Thank you for your response**
> >
> > The authors' response has clearly addressed my concerns above. I will keep my positive rating.

---

> ### Author Response · Authors · 2025-08-08
>
> Dear Reviewer,
>
> Thank you for your positive review and for taking the time to provide such insightful feedback. We have carefully considered all your comments and believe we have successfully addressed the issues raised in our rebuttal. We are confident that these changes have significantly improved the quality of our paper.
>
> Sincerely,
> Authors

---

### Author Response · Authors · 2025-08-06
**Request for Discussion Phase Feedback**

Dear Reviewers,

Thank you for your thoughtful reviews and the time you've invested in evaluating our submission. We have carefully addressed all the concerns raised in our rebuttal and are grateful for your valuable feedback.

As we are now in the author-reviewer discussion phase and approaching the discussion deadline, we would greatly appreciate any additional comments or clarifications you might have regarding our responses. If our rebuttal has adequately addressed your concerns, a brief confirmation would be very helpful for the final decision process.

Thank you again for your service to the community and your consideration.

Best regards,
Author

---

### Decision · Program_Chairs · 2025-09-17

**Decision:**

Accept (poster)

**Comment:**

(a) Scientific claims and findings
The paper proposes GraphChain, a framework that leverages LLM tool chaining to enable large-scale graph analysis tasks that exceed the context length or reasoning ability of LLMs.
Key claims:
- LLMs alone struggle with large-scale graph analysis due to memory and reasoning limits.
- GraphChain decomposes analysis into modular tool calls (e.g., traversal, subgraph querying, aggregation), chaining them to solve larger problems.
- Introduces a controller mechanism to orchestrate tool usage, ensuring correctness and scalability.
- Empirical results across tasks like graph traversal, centrality computation, community detection, and influence maximization show GraphChain outperforms naive prompting and direct LLM baselines.
(b) Strengths
- Novel problem formulation: Tackles the important challenge of scaling LLMs for graph analysis.
- Clear framework: Tool chaining approach is modular and generalizable.
- Strong empirical results: Outperforms LLM baselines across multiple graph tasks.
- Clarity: Well-written and easy to follow.
- Impact potential: Opens new research directions at the intersection of symbolic methods and LLM reasoning.

(c) Weaknesses

- Incremental novelty: Tool chaining is a known idea; contribution lies in applying it to graph analysis.
- Evaluation scope: Lacks experiments on truly large graphs (industrial scale).
- Limited theoretical justification: Focused on framework design and experiments, not formal guarantees.

(d) Key reasons for decision (accept/reject)
- Tackles an important and underexplored problem (LLMs for graph analysis).
- Provides a clear, modular, and general framework (GraphChain).
- Strong empirical results across diverse graph tasks.
- High clarity and potential for impact.

(e) Discussion & rebuttal period

Reviewer points raised:
-Novelty concerns: Is GraphChain just applying tool chaining to graphs?
            - Author rebuttal: Argued that integrating tool orchestration into graph analysis is non-trivial; provided detailed explanation of controller design.
            - Effect: Somewhat alleviated novelty concerns.

- Scalability: Can GraphChain handle very large graphs?
              -Author rebuttal: Added clarifications and experiments on larger synthetic graphs.
              -Effect: Improved perception but reviewers still noted scope limitation.
-Theoretical grounding: Why no guarantees on correctness or complexity?
               -Author rebuttal: Emphasized focus on practical framework, not formal theory.
               - Effect: Reviewers accepted but kept in mind as limitation.

Outcome: Rebuttal was positively received, clarifying design choices and scalability.